# MHLA: Restoring Expressivity of Linear Attention via Token-Level Multi-Head

**Kewei Zhang**[1*], **Ye Huang**[1*], **Yufan Deng**[1], **Jincheng Yu**[2], **Junsong Chen**[2],
**Huan Ling**[2], **Enze Xie**[2], **Daquan Zhou**[1†]
[1]Peking University, Shenzhen Graduate School    [2]NVIDIA
Project Page:   dagroup-pku.github.io/MHLA

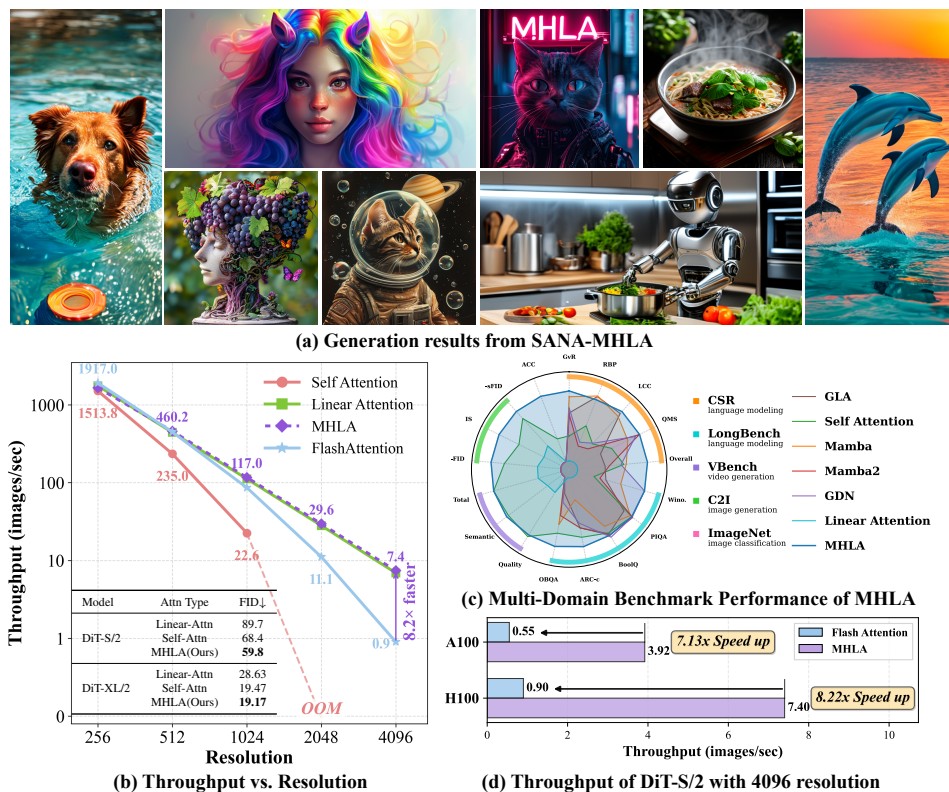

Figure 1: (a) **Generation results from our fine-tuned SANA model using MHLA.** (b) **Performance and efficiency comparison between the proposed MHLA and baselines.** The throughput was tested on the NVIDIA H100 Tensor Core GPU. Following the previous method, we report the FID in the table at a resolution of $256 \times 256$. (c) **Average rank and entropy of attention scores** for DeiT-T with different attention types, showing MHLA yields richer and more focused attention. (d) **Throughput of DiT-S/2 at 4096 resolution across different devices.** *All improvements are solely due to MHLA*, and can be further combined with orthogonal techniques for even greater speedups.

## Abstract

While the Transformer architecture dominates many fields, its quadratic self-attention complexity hinders its use in large-scale applications. **Linear attention** offers an efficient alternative, but its direct application often degrades performance, with existing fixes typically re-introducing computational overhead through extra modules (e.g., depthwise separable convolution) that defeat the original purpose. In this work, we identify a key failure mode in these methods: **global context collapse**, where the model loses representational diversity. To address this, we propose **Multi-Head Linear Attention (MHLA)**, which preserves this diversity by computing attention within divided heads along the token dimension. We prove that MHLA maintains linear complexity while recovering much of the

---

∗ Equal contribution.

† Correspond to daquan.zhou@pku.edu.cn

expressive power of softmax attention, and verify its effectiveness across multiple domains, achieving a **3.6%** improvement on ImageNet classification, a **6.3%** gain on NLP, a **12.6%** improvement on image generation, and a **41%** enhancement on video generation under the same time complexity.

# 1 INTRODUCTION

Self-attention is the core module for the recent dominant model architecture, Transformer, for both computer vision (Dosovitskiy et al., 2021), natural language processing (Vaswani et al., 2017), and generative tasks (Rombach et al., 2022). However, its quadratic time and memory complexity severely limit its scalability to long sequence tasks such as high-resolution image generative and video generation tasks (Zhou et al., 2022; Kong et al., 2024; Zhou et al., 2024).

To address the efficiency issue, a growing line of research (Katharopoulos et al., 2020; Choromanski et al., 2021) has developed linear attention mechanisms that replace the softmax kernel with associative feature maps. These approaches reduce the computational and memory complexity of attention from quadratic to linear by compressing all keys and values into a global summary. Although this improves efficiency, it eliminates one of the key advantages of softmax attention—its ability to adapt to each query individually. Consequently, linear attention often experiences notable accuracy degradation, particularly in long-sequence modeling tasks.

Recent works (Fan et al., 2025b; Han et al., 2023; 2024) have sought to mitigate the performance degradation of linear attention by integrating components such as depthwise convolutions and gating modules. However, this reliance on external modules introduces additional computational overhead and continues to suffer from performance degradation as sequence length increases. In this paper, we present a solution to the performance bottleneck in linear attention that requires no additional depthwise convolution or self-attention modules. Our key insight is that, in conventional linear attention design, all tokens are compressed into a single global key–value summary (KV summary) that is shared by every query. This design could have reduced the model's representation capacity, as illustrated in Figs. 1b and 2. To evaluate diversity, we compare the rank of the attention weight matrices across different models. We find that using a shared global KV summary limits the model's capacity to represent rich interactions, effectively capping it at a fixed rank. As sequences grow longer, this constraint tends to push the attention weights toward a more uniform distribution. In practice, this reduces diversity and degrades performance on tasks where queries must concentrate on a small subset of relevant tokens.

Our design goal is therefore simple: restore query-dependent diversity, the ability for different queries to retrieve different contexts, without sacrificing linear-time behavior or introducing heavy auxiliary modules.

Thus, we introduce Multi-head Linear Attention (MHLA) to achieve the aforementioned characteristics. Specifically, MHLA partitions tokens into non-overlapping blocks ("heads" in the spatial dimension), computes local key-value summaries, and lets each query block compute a query-conditioned mixture over these summaries to retrieve a tailored context; within the selected blocks, token contributions are further refined by a query-dependent reweighting module. Thanks to the simplicity of MHLA, the implementation only relies on standard GEMMs, keeping the overall computational overhead negligible with $O(N)$ complexity, retaining compatibility with streaming/stateful execution. It was clearly observed that adding MHLA raise the rank of the attention weights matrix significantly, as shown in Fig. 3b. The difference between previous linear attentions and MHLA is briefly illustrated in Fig. 2.

We validate MHLA on image classification, image generation, natural language processing, and video generation tasks. Experiments show that MHLA consistently outperforms existing linear attention baselines with negligible computational overhead. Our main contributions are summarized as follows:

- We conduct an in-depth analysis of linear attention and identify one of the root causes of its performance degradation: the absence of grouping along the token dimension during similarity calculation. This limitation can be quantified by examining the rank of the attention matrix.

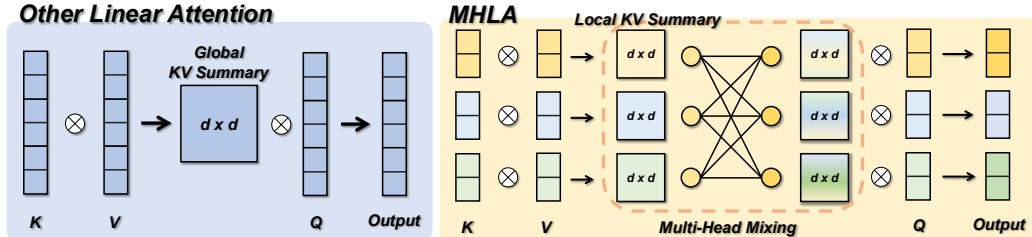

Figure 2: **Comparison between the proposed MHLA and other linear attentions.** MHLA divides multiple heads on the token dimension. Through Multi-Head Mixing, MHLA restores query-conditioned selectivity by mixing KV summaries with query-specific weight, improving token-level diversity while keeping linear complexity.

- We propose a new formulation of linear attention that achieves state-of-the-art performance on both discriminative and generative tasks, while maintaining $O(N)$ computational complexity and avoiding reliance on additional modules.
- We conduct extensive experiments across various tasks, achieving state-of-the-art performance. On ImageNet, MHLA delivers a **3.6%** accuracy gain over self-attention, while on image generation tasks it improves the performance of the DiT architecture by **12.6%**. MHLA also achieves a **6.3%** improvement on natural language processing tasks and provides a substantial **41%** improvement compared to vanilla linear attention in video generation tasks.

## 2 RELATED WORKS

Transformers (Vaswani et al., 2017) have advanced various fields (Devlin et al., 2019; Dosovitskiy et al., 2021; Saharia et al., 2022), but their quadratic time and memory complexity due to self-attention limit scalability, especially for long sequences. To overcome this, linear attention mechanisms (Katharopoulos et al., 2020; Choromanski et al., 2021) have been proposed, which replace softmax with kernel-based methods to achieve linear time complexity. While these mechanisms improve the efficiency, they often lose expressiveness, making them suffer from a performance drop in capturing complex token interactions. Several solutions (Fan et al., 2025b; Han et al., 2023), including adding convolutional layers or gating mechanisms, have attempted to recover performance but tend to introduce additional computational costs. See the detailed related works in the Appendix A.

## 3 ANALYSIS OF LINEAR ATTENTION

### 3.1 PRELIMINARY

We first formulate the calculation of the attention weights for both self-attention and linear attention mechanism. Given an input token sequence $X \in \mathbb{R}^{N \times d}$, we first compute queries, keys, and values via $Q = XW_Q$, $K = XW_K$, $V = XW_V$, where $W_Q, W_K, W_V \in \mathbb{R}^{d \times d}$ are learnable projections. The attention output of the token $i$ can be expressed as:

$$Y_i = \frac{\sum_{j=1}^{N} \text{Sim}(Q_i, K_j)V_j}{\sum_{m=1}^{N} \text{Sim}(Q_i, K_m)}, \tag{1}$$

where $\text{Sim}(\cdot, \cdot)$ calculates the similarity between the input matrix. In softmax attention (Vaswani et al., 2017), $\text{Sim}(Q_i, K_j) = \exp(Q_i K_j^\top / \sqrt{d})$, all pairwise similarities need to be calculated and normalized per query, resulting in $O(N^2)$ complexity.

Linear attention replaces the exponential kernel with a positive feature map $\phi(\cdot)$ such that

$$\text{Sim}(Q_i, K_j) \approx \phi(Q_i)\phi(K_j)^\top, \qquad Y_i = \frac{\phi(Q_i)\left(\sum_{j=1}^{N} \phi(K_j)^\top V_j\right)}{\phi(Q_i)\left(\sum_{m=1}^{N} \phi(K_m)^\top\right)}, \tag{2}$$

where the numerator and denominator can be precomputed as a global key–value summary $G = \sum_j \phi(K_j)^\top V_j$ and normalizer $z = \sum_m \phi(K_m)^\top$, respectively. This reduces the complexity from $O(N^2)$ to $O(Nd_\phi)$, enabling linear-time scaling with sequence length.

## 3.2 GLOBAL CONTEXT COLLAPSE

Linear attention achieves linear-time complexity by reusing a global key–value summary across all queries, which can be formulated as $G = \sum_{j=1}^{N} \phi(K_j)^\top V_j \in \mathbb{R}^{d \times d}$. But this fixed-size design introduces an intrinsic information bottleneck:

> **Observation**
>
> As the sequence length $N$ increases, the information requiring representation exceeds the capacity of the fixed-size $d \times d$ matrix, leading to performance saturation. We term this phenomenon *global context collapse*.

This observation can be quantified using two complementary metrics, which are the rank and the sparsity of the attention matrix:

**Rank limitation.** The rank of the attention matrix has been widely studied as a key indicator of feature diversity and representational capacity in attention mechanisms (Fan et al., 2025b; Han et al., 2023; Bhojanapalli et al., 2020). Specifically, with $\widetilde{Q} = \phi(Q)$ and $\widetilde{K} = \phi(K)$, global linear attention produces

$$A_{\text{lin}} = \widetilde{Q}\,\widetilde{K}^\top \in \mathbb{R}^{n \times n}, \qquad \text{rank}(A_{\text{lin}}) \leq \min\{\text{rank}(\widetilde{Q}), \text{rank}(\widetilde{K})\} \leq d.$$

> **Conclusion 1**
>
> Regardless of $N$, the representational capacity of $A_{\text{lin}}$ is strictly bounded by $d$. Although several prior studies have attempted to increase the rank of Key–Value summaries (Fan et al., 2025b; Cao & Wang, 2025), this bound results in a severely rank-deficient approximation of the full $n \times n$ attention matrix when $n \gg d$, constraining the model's ability to capture diverse, query-conditioned attention patterns.

We empirically verify this effect in Fig. 3b, which shows that the rank of attention scores in linear-attention-based models is consistently capped by the head dimension (typically $d_h \leq 72$), and the relative expressivity of the attention map degrades as the sequence length increases.

**Loss of sparsity.** The sparsity of the attention matrix is a critical factor influencing the performance of attention mechanisms. Sparse distributions generally exhibit lower entropy, concentrating probability mass on a smaller set of informative tokens (Zhang et al., 2025; Deng et al., 2023), which benefits model optimization. Linear attention, however, computes scores by first compressing all key–value pairs into a single global summary, and each query interacts with this shared representation only once. In contrast, softmax attention leverages the exponential function to enable each query $q_i$ to produce a distinct distribution over tokens (see Appendix B). Because linear attention relies on the same aggregated representation for all queries, it cannot reweight individual keys according to query-specific relevance.

> **Conclusion 2**
>
> As the sequence length $N$ increases, the contribution of each token becomes negligible. Consequently, the attention weight distribution approaches uniformity, reducing the sparsity and impairing the model's ability to selectively emphasize informative tokens.

To quantify this effect, we compute the average entropy of the attention scores over 500 random samples for each attention variant. For each row of the attention score matrix, lower entropy indicates that the distribution is closer to a one-hot vector, reflecting stronger concentration on a single token. As shown in Fig. 3a and Fig. 3b, linear attention exhibits significantly higher entropy, confirming its lack of focus compared to softmax-based attention.

Taken together, these findings reveal that the reliance on a single global key–value summary in linear attention leads to a severe collapse in representational capacity, manifested as both rank deficiency and elevated entropy in the attention map. We refer to this phenomenon as *global context*

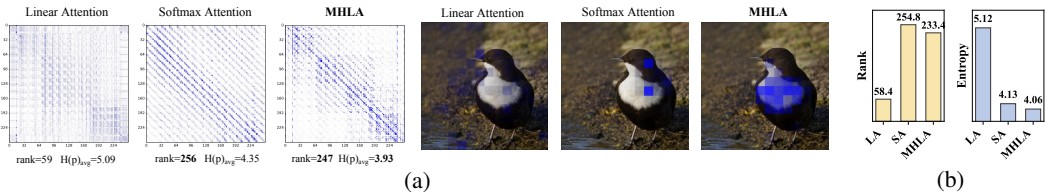

Figure 3: (a) **Visualization of attention score and attention maps** of MHLA and baselines. (b) **Average rank and entropy** of attention scores for DeiT-T, showing MHLA yields richer and more focused attention.

*collapse*. Fig. 3a visualizes attention scores and maps, clearly illustrating the inability of linear attention to capture fine-grained information. This observation motivates the development of methods that restore query-conditioned token-level diversity while preserving the linear-time complexity of the attention mechanism, which was detailed in the next section.

# 4 MULTI-HEAD LINEAR ATTENTION

## 4.1 OVERVIEW

Here we formalize the proposed **Multi-Head Linear Attention (MHLA)**. As shown in Fig. 4a. MHLA operates by splitting the sequence along the token dimension into multiple "heads" and running linear attention in parallel across these "heads". Let the input sequence be $X \in \mathbb{R}^{N \times d}$, projected to queries, keys, and values: $Q = XW_Q, \quad K = XW_K, \quad V = XW_V$, with $Q, K, V \in \mathbb{R}^{N \times d}$. For efficiency, we adopt a kernelized formulation, denoting $\widetilde{Q} = \phi(Q), \widetilde{K} = \phi(K)$ for a chosen feature map $\phi(\cdot)$.

Standard linear attention aggregates all tokens into a single global $d \times d$ summary shared by every query, which reduces expressivity by collapsing token-level diversity. To mitigate this, we split the sequence into $M$ non-overlapping blocks (the MHLA "heads"), with block $b$ containing $N_b$ tokens and $\sum_{b=1}^{M} N_b = N$. In practice on vision models, blocks are defined on spatial (2D) or spatiotemporal (3D) grids rather than by flattening to 1D. For each block $b$ we compute a local key–value summary and its normalizer:

$$S_b = \sum_{j \in b} \widetilde{K}_j V_j^\top \in \mathbb{R}^{d \times d}, \qquad z_b = \sum_{j \in b} \widetilde{K}_j \in \mathbb{R}^d. \tag{3}$$

To restore query adaptivity, **MHLA** constructs a distinct mixture of all key–value summaries for each query block $i$ through *Multi-Head Mixing*. Queries in block i can then attend to this mixture, where different key–value summaries are weighted according to the attention preferences of the current query block. Let $m_i \in \mathbb{R}^M$ denote the nonnegative, learnable mixing coefficients for block i, which are optimized during training. The mixed summaries are then defined as $\widetilde{S}_i = \sum_{b=1}^{M} m_{i,b} S_b$, and the corresponding normalizer is $\widetilde{z}_i = \sum_{b=1}^{M} m_{i,b} z_b$.

The process can be done with a highly hardware-efficient GEMM operation between key–value summaries and coefficient matrix $\mathcal{M}_c \in \mathbb{R}^{M \times M}$ consisting of $m_i$. Given a query vector $\widetilde{q} \in \mathbb{R}^d$ from block $i$, the output is

$$o = \frac{\widetilde{q}^\top \widetilde{S}_i}{\widetilde{q}^\top \widetilde{z}_i} = \frac{\sum_{b=1}^{M} m_{i,b} \widetilde{q}^\top S_b}{\sum_{b=1}^{M} m_{i,b} \widetilde{q}^\top z_b}. \tag{4}$$

Each output element can thus be interpreted as a query-specific, block-dependent recombination of the entire value sequence. In tasks like language modeling and video generation, the normalizer term can be omitted for better training stability (Qin et al., 2022) when the sequence is getting longer.

## 4.2 MULTI-HEAD MIXING

The core of MHLA's adaptivity is a learned coefficient matrix $\mathcal{M}_c \in \mathbb{R}^{M \times M}$. The element at position $(i, j)$ denotes the affinity between query-block $i$ and the local key–value summary of block $j$. Equivalently, the $i$-th row of $\mathcal{M}_c$, denoted $m_i$, specifies how query-block $i$ linearly combines the $M$ local summaries into a query-specific global summary.

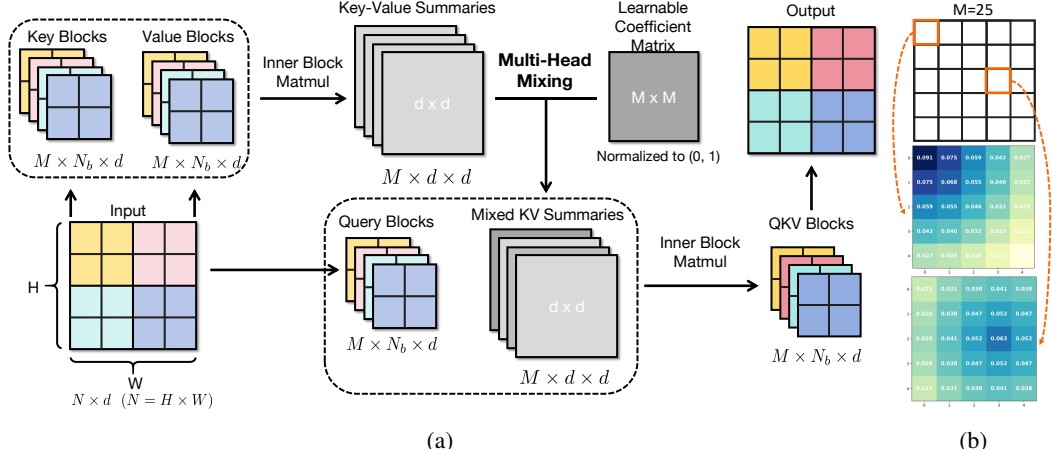

Figure 4: (a) **Overview of the proposed Multi-Head Linear Attention.** (b) We visualize two rows of the initialized Learnable Coefficient Matrix corresponding to $Block\ 1$ and $Block\ 14$ separately when M is 25. We reshape the two rows and the M dimension in 2D for better understanding.

Each row $m_i$ is produced and learned end-to-end; in practice we enforce nonnegativity and normalization. Because blocks are defined along spatial or spatiotemporal axes, we initialize $\mathcal{M}_c$ to favor locality: for row $i$ we set initial coefficients as $m_{i,j}^{(0)} \propto 1 - \text{dist}(i,j) / \max_k(\text{dist}(i,k))$, where $\text{dist}(i,j)$ measures the Euclidean distance and $\max_k \text{dist}(i,k)$ is the maximum distance from i to any position k. The coefficients are then normalized such that $\sum_j m_{i,j}^{(0)} = 1$. A visualization of this initialization can be found in Fig. 4b. This locality-biased initialization produces more stable and faster convergence while leaving $\mathcal{M}_c$ free to adapt during training. To further ensure stability, we clip the coefficients to the interval (0, 1) on every update.

The token-level effect of the Multi-Head Mixing is transparent. Let $b(t)$ denote the block index of token $t$. Writing each local summary as a sum over its tokens, $G_j = \sum_{t \in \text{block } j} \widetilde{K}_t V_t^\top$, the mixture for query-block $i$ expands to

$$\widetilde{S}_i = \sum_{j=1}^{M} m_{i,j} S_j = \sum_{t=1}^{N} m_{i,b(t)} \widetilde{K}_t V_t^\top \in \mathbb{R}^{d \times d}.$$

For a query vector $\widetilde{q} = \phi(q)$ (from block $i$), the numerator of the kernelized update becomes

$$\widetilde{q}^\top \widetilde{S}_i = \sum_{t=1}^{N} m_{i,b(t)} \big(\widetilde{q}^\top \widetilde{K}_t\big) V_t^\top \in \mathbb{R}^d. \tag{5}$$

Eq. 5 makes the mechanism transparent: each query-block rescales the contribution of entire blocks via $m_i$, and within each block the usual kernel inner product $\widetilde{q}^\top \widetilde{K}_t$ differentiates tokens. Thus, MHLA restores *query-conditioned, token-level* weighting in a two-stage manner (block selection × intra-block reweighting). Importantly, all operations reduce to blockwise summary computation and linear combinations of $M$ matrices of size $d \times d$, so asymptotic complexity remains linear in $N$ while expressive capacity is substantially increased.

**Chunkwise parallel form of MHLA.** Linear attention commonly employs *chunkwise parallel training* (Hua et al., 2022; Sun et al., 2023) to maintain linear-time complexity under causal masking, by partitioning the sequence into blocks and updating a running summary per block. MHLA naturally fits this setting: each head can be directly mapped to a chunk, and we maintain one local summary $S_b$ per chunk. For a detailed derivation, see Appendix C.

### 4.3 ANALYSIS OF MULTI-HEAD LINEAR ATTENTION

**Rank analysis.** Partition the sequence into $M$ non-overlapping blocks of size $N_b$. Let the query matrix be $\widetilde{Q} = [\widetilde{Q}_1^\top, \ldots, \widetilde{Q}_M^\top]^\top$ with $\widetilde{Q}_b \in \mathbb{R}^{n_b \times d}$. From Eq. 5, in the calculation of attention score,

Table 1: **Comparison between Self Attention, Linear Attention, and MHLA.** We report computation complexity, maximum achievable rank, memory complexity and query-conditioned selectivity.

| Method | Time Complexity | Rank Bound | Memory Complexity | Query-Conditioned |
|---|---|---|---|---|
| Self Attention | $O(N^2 d)$ | $N$ | $O(N^2)$ or $O(N)$ | ✓ |
| Linear Attention | $O(N d^2)$ | $d$ | $O(d^2)$ | ✗ |
| MHLA (ours) | $O(N d^2 + M^2 d^2)$ | $\sum_{b=1}^{M} \min(n_b, d)$ | $O(M d^2)$ | ✓ |

the mixed key sequence seen by query-block $i$ can be expressed as

$$Y_i = \big[\, m_{i,b(1)} k_1,\ m_{i,b(2)} k_2,\ \ldots,\ m_{i,b(n)} k_n \,\big] \in \mathbb{R}^{d \times n},$$

where $m_{i,b(t)}$ is the mixing coefficient selecting the block of token $t$. The attention submatrix contributed by query-block $i$ is $A_i = \widetilde{Q}_i Y_i \in \mathbb{R}^{N_b \times N}$, and the full attention matrix is $A_{\mathrm{MHLA}} = \big[A_1\ A_2\ \cdots\ A_M\big]^\top \in \mathbb{R}^{n \times n}$. Then applying standard rank inequalities gives

$$\mathrm{rank}(A_b) \ \leq\ \min\big\{\mathrm{rank}(\widetilde{Q}_b), \mathrm{rank}(Y_b)\big\} \ \leq\ \min(n_b, d),$$

which yields the global bound $\mathrm{rank}(A_{\mathrm{MHLA}}) \ \leq\ \min\big(n,\ \sum_{b=1}^{M} \min(n_b, d)\big)$.

This upper bound is *attainable* under mild, generic conditions: if each block product $\widetilde{Q}_b Y_b$ has full row rank $r_b = \min(n_b, d)$ and the row spaces of $\{\widetilde{Q}_b Y_b\}_{b=1}^{M}$ are linearly independent, then we get $\mathrm{rank}(A_{\mathrm{MHLA}}) = \min(n,\ \sum_{b=1}^{M} r_b)$. Even when the independence assumption is not fully satisfied, the blockwise mixture still expands the diversity of the row spaces, causing $\mathrm{rank}(A_{\mathrm{MHLA}})$ to grow roughly additively with $M$. We empirically validate this behavior in Fig. 3b, where MHLA consistently achieves a substantially higher attention-score rank than other linear attention variants— and does so *without* relying on auxiliary components such as depth-wise convolutions. This confirms that MHLA natively restores much of the representational capacity lost in global linear attention, whose rank remains strictly limited by $d$ regardless of the sequence length $N$.

**Sparsity analysis.** The learned coefficient matrix $\mathcal{M}_c$ allows each query-block to assign higher weights to a subset of blocks that are more relevant, effectively pruning irrelevant tokens at the block level. Within each selected block, the kernel inner products $\widetilde{q}^\top \widetilde{K}_t$ further differentiate token contributions, leading to sharper and more concentrated attention distributions. We validate this effect empirically in Fig. 3b, where MHLA consistently yields lower attention entropy compared to other linear-attention baselines and even the softmax attention. This confirms that MHLA preserves query-conditioned selectivity and achieves substantially higher sparsity, enabling the model to attend to a small, semantically relevant subset of tokens rather than spreading attention uniformly.

**Efficiency analysis.** The computation of MHLA consists of local Key–value summary computation, Multi-Head Mixing, and output computation, with a time complexity of $O\big(M N_b d^2 + M^2 d^2 + M N_b d^2\big) = O(N d^2 + M^2 d^2)$. To better capture local information while ensuring efficiency, the number of blocks $M$ is usually set to satisfy $M^2 \leq N$. Therefore, $N d^2$ becomes the leading term and the time complexity of MHLA is $O(N d^2)$. The comparison of self attention, linear attention, and MHLA is summarized in Tab. 1. We also provide an empirical analysis of the scaling relationship between N and M in Appendix F.4 that verifies the induced complexity.

## 5 EXPERIMENTS

### 5.1 IMAGE CLASSIFICATION

**Settings.** We adopt the training configurations from prior work (Fan et al., 2025b;a; Touvron et al., 2021). The proposed MHLA is integrated into two representative architectures, DeiT (Touvron et al., 2021) and VLT (Fan et al., 2025b), across multiple model scales. The models are trained on ImageNet-1K (Deng et al., 2009). For VLT, we strictly follow the setup in (Fan et al., 2025b). All models are trained for 300 epochs with a batch size of 1024 and a peak learning rate of 1e-3. The head number $M$ is set to 16 if there is no extra description. See Appendix E for more details.

**Results.** We evaluate the pretrained DeiT models described above and report the result in Tab. 2a, which clearly shows the superior performance of the proposed MHLA. We reach the best accuracy

Table 2: **Comparison on Image Classification task.** MHLA achieves the best accuracy with minimal parameter overhead on DeiT models, and outperforms **Transformer**-, **LA**-, and **Mamba**-based SOTAs. Results marked with an * are reproduced under the same training setup as MHLA-VLT.

(a) Comparison of different attentions on DeiT.

| Attention Type | Params | FLOPs | Top1-ACC |
|---|---|---|---|
| Comparison on Deit-T Setting | | | |
| Self Attn | 5.7M | 1.1G | 72.2 |
| Linear Attn | 5.7M | 1.1G | 69.8 |
| Focused LA (Han et al., 2023) | 6.1M | 1.1G | 74.1 |
| Inline Attn (Han et al., 2024) | 6.5M | 1.1G | 74.5 |
| MALA (Fan et al., 2025a) | 6.3M | 1.1G | 75.1 |
| **MHLA (Ours)** | **5.7M** | 1.1G | **75.8** |
| Comparison on Deit-S Setting | | | |
| Self Attn | 22M | 4.2G | 79.8 |
| Linear Attn | 22M | 4.2G | 77.6 |
| RALA (Fan et al., 2025b) | 24M | 4.6G | 80.4 |
| MALA (Fan et al., 2025a) | 24M | 4.6G | 80.3 |
| **MHLA (Ours)** | **22M** | 4.2G | **81.0** |

(b) Comparison with SOTA models on ImageNet-1K.

| Cost | Model | Params | FLOPs | Top1-ACC |
|---|---|---|---|---|
| ~2.5G | FL-PVT-T (Han et al., 2023) | 12M | 2.0G | 77.8 |
| | FL-PVTv2-B1 (Han et al., 2023) | 13M | 2.2G | 79.5 |
| | MSVMamba-M (Shi et al., 2024) | 12M | 1.5G | 79.8 |
| | NAT-M (Hassani et al., 2023) | 20M | 2.7G | 81.8 |
| | RAVLT-T (Fan et al., 2025b) | 15M | 2.4G | 82.3* |
| | MAViT-T (Fan et al., 2025a) | 16M | 2.5G | 82.4* |
| | **MHLA-VLT-T** | 16M | 2.4G | **82.6** |
| ~4.5G | FAT-B3 (Fan et al., 2023) | 29M | 4.4G | 83.6 |
| | Vmamba-T (Liu et al., 2024) | 30M | 4.9G | 82.6 |
| | MV-T (Hatamizadeh & Kautz, 2025) | 32M | 4.4G | 82.3 |
| | MSVMamba-T (Shi et al., 2024) | 32M | 5.1G | 83.0 |
| | MAViT-S (Fan et al., 2025a) | 27M | 4.6G | 84.3* |
| | **MHLA-VLT-S** | 27M | 4.6G | **84.6** |

in linear attention across all model sizes, while introducing the fewest extra parameters compared with baselines. We then port the proposed MHLA to VLT (Fan et al., 2025b) and evaluate the performance under the same settings. The results are shown in Tab. 2b, illustrating the proposed MHLA's state-of-the-art performance with consistent improvements compared with baseline models.

## 5.2 IMAGE GENERATION

**Settings.** *1) For Class-to-Image(C2I) generation*, we train DiT (Peebles & Xie, 2023) and DiG (Zhu et al., 2025) from scratch for 400k steps on ImageNet-1K (Deng et al., 2009) with batch size 256 and learning rate 1e-4, following their original settings. We evaluate five variants in DiT and DiG, where the original self-attention (DiT) or GLA (Yang et al., 2024) (DiG) is replaced by our MHLA while keeping other components unchanged. The head number is set to 16 for both 256 and 512 resolutions. We try extra CPE (Chu et al., 2021) and the output gating module (Yang et al., 2024). Their effects are analyzed in Appendix F.2. *2) For Text-to-Image(T2I) generation*, we finetune a Sana-0.6B (Xie et al., 2024) model from official checkpoint. Both the original linear attention and our MHLA variant are trained for 40k steps with a batch size of 256.

**C2I results.** The main quantitative results are summarized in Tab. 3a, where our method consistently achieves state-of-the-art performance across all DiT model sizes. In addition, Fig. 1b compares the throughput of our MHLA with baseline attention mechanisms on DiT-S as the input resolution increases. Notably, MHLA maintains throughput nearly identical to linear attention while delivering performance on par with, or even surpassing, self-attention. At 512 resolution, MHLA achieves better FID scores while doubling the throughput of self-attention. To fur-

Table 3: **Class-to-Image Generation.** Across all model sizes, MHLA achieves the best performance. Notably, at L and XL scales, it matches self-attention performance without relying on any extra modules.

(a) Comparison of attention types across models.

| Model | Attention Type | Resolution | FID ↓ |
|---|---|---|---|
| DiT-S/2 | Self Attention | 256 | 68.40 |
| | Linear Attention | 256 | 89.72 |
| | MHLA (Ours) | 256 | **59.80** |
| | Self Attention | 512 | 84.54 |
| | Linear Attention | 512 | 125.33 |
| | MHLA (Ours) | 512 | **78.63** |
| DiG-S/2 | GLA (Yang et al., 2024) | 256 | 62.06 |
| | GLA | 512 | 99.04 |
| | MHLA (Ours) | 256 | **59.49** |
| DiT-B/2 | Self Attention | 256 | 43.47 |
| | Linear Attention | 256 | 60.47 |
| | MHLA (Ours) | 256 | **37.47** |
| DiT-L/2 | Self Attention | 256 | 23.33 |
| | Linear Attention | 256 | 32.35 |
| | MHLA (Ours, w/None) | 256 | 25.37 |
| | MHLA (Ours, w/ CPE) | 256 | 24.21 |
| | MHLA (Ours, w/ CPE+Gating) | 256 | **21.37** |
| DiT-XL/2 | Self Attention | 256 | 19.47 |
| | Linear Attention | 256 | 28.63 |
| | MHLA (Ours, w/ None) | 256 | *20.32* |
| | MHLA (Ours, w/ CPE) | 256 | 22.79 |
| | MHLA (Ours, w/ CPE+Gating) | 256 | **19.17** |

(b) Fast adaptation results on DiT-XL/2.

| Model | Attention Type | FID ↓ | IS ↑ | sFID ↓ |
|---|---|---|---|---|
| DiT-XL/2 | Self Attention | 9.62 | 121.50 | 6.85 |
| | MHLA (Ours) | **8.34** | 121.27 | **5.52** |
| DiT-XL/2(G) | Self Attention | 2.27 | 278.24 | 4.60 |
| | MHLA (Ours) | 2.54 | 252.07 | 4.67 |

ther demonstrate the fast adaptation ability of our approach to existing models, we fine-tune the pretrained DiT-XL/2 model for 400k steps under the same settings. As shown in Tab. 3b, our model achieves a lower FID score than DiT-XL/2 without CFG, and delivers comparable performance when CFG is applied. Full results of the experiments can be found in Appendix F.

**Analysis.** Although we add modules such as DWConv (CPE) (Fan et al., 2025b) to smaller DiT models, it is worth noting that their benefits diminish as model size increases (CPE even degrades performance on DiT-XL). As shown in Tab. 3a, plain MHLA already matches the performance of self-attention on XL models, while adding CPE leads to regression. These results highlight the intrinsic advantage of MHLA and suggest that, although modules like DWConv may offer gains at small scales, their benefits do not scale with model size or sequence length.

Table 4: Comparison on T2I models.

| Model | FID↓ | CLIP ↑ | GenEval ↑ |
|---|---|---|---|
| PixArt-$\alpha$ (Chen et al., 2023) | 6.14 | 27.55 | 0.48 |
| PixArt-$\Sigma$ (Chen et al., 2024) | 6.34 | 27.62 | 0.52 |
| SANA* (Xie et al., 2024) | 6.10 | 28.15 | 0.64 |
| SANA-MHLA | **5.90** | **28.26** | **0.68** |

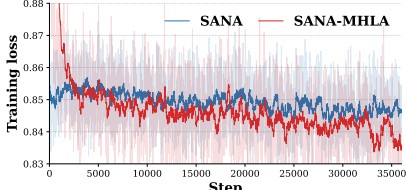

Figure 5: Loss comparison.

**Fast adaptation to SANA.** As shown in Tab. 4, replacing linear attention with MHLA consistently improves multiple evaluation metrics, surpassing not only the baseline Sana model but also the PixArt (Chen et al., 2023) series. Fig. 5 further visualizes the training loss curves. The MHLA-based model rapidly adapts, matching the pretrained checkpoint within the first 2k steps and subsequently converging to a lower loss. This demonstrates MHLA's fast adaptation capability and promising performance at a larger model scale.

## 5.3 VIDEO GENERATION

Video generation involves **extremely long sequence lengths**, where quadratic attention becomes prohibitively slow. To evaluate MHLA under such ultra-long contexts, we fine-tune a pretrained `Wan2.1-1.3B` model by replacing its Flash Attention modules with MHLA. For comparison, we also fine-tune a version with full vanilla linear attention (LA). The training uses 81-frame videos at 480×800 resolution, corresponding to a sequence length of **31,500 tokens**, with the mixing-head number $M = 105$. In addition, we train a hybrid model with 2/3 of the layers replaced with MHLA.

We evaluate all models on VBench (Tab. 5). MHLA achieves **substantially stronger performance** than vanilla LA *at the same latency*. Under such extreme sequence lengths, vanilla LA suffers from *global context collapse*, leading to severe degradation, whereas MHLA maintains linear-time complexity and recovers performance comparable to the original FlashAttention-based Wan2.1-1.3B, achieving a **2.1×** **inference speedup**. The hybrid model further offers an excellent trade-off, delivering a **1.6× speedup** with even better overall performance. As shown in Fig. 6, MHLA also adapts rapidly during fine-tuning, quickly approaching the pretrained model's loss trajectory. In contrast, vanilla LA fails to train under ultra-long sequences, with its loss plateauing at a high level.

Table 5: MHLA in Video Generation. Wan-FA indicates a pretrained Wan2.1-1.3B. Wan-MHLA and Wan-LA replace all layers with MHLA and Linear Attention, respectively. Wan-MHLA-H only replaces 2/3 layers.

| Model | Quality ↑ | Semantic ↑ | Total ↑ | Latency (s) ↓ |
|---|---|---|---|---|
| Wan-FA | **85.23** | 75.65 | 83.31 | 166 |
| Wan-LA | 69.96 | 11.38 | 58.24 | 82 |
| Wan-MHLA | 84.26 | 76.16 | 82.62 | **81** |
| Wan-MHLA-H | 84.87 | **79.59** | **83.82** | 103 |

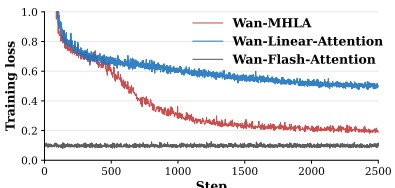

Figure 6: Loss comparison on Wan-2.1-1.3B. MHLA shows a much stronger convergence capability.

## 5.4 NATURAL LANGUAGE PROCESSING

To evaluate MHLA under autoregressive modeling, we test its performance in language modeling. Following GLA (Yang et al., 2024), we train a 0.3B model from scratch on 10B tokens from FineWeb-Edu (Penedo et al., 2024) with a batch size of 0.25M tokens, using a cosine learning rate schedule (max LR 3e-4), weight decay of $0.01$, and gradient clipping of $1.0$. The head number $M$ is set to 32 for MHLA with a training context length of 2048.

In Tab. 6, we report language modeling perplexity and zero-shot commonsense reasoning accuracy, where MHLA achieves performance comparable to Transformer++(Touvron et al., 2023) and state-of-the-art linear models such as GDN (Yang et al., 2025) and Mamba2 (Dao & Gu, 2024). On

LongBench (Bai et al., 2024) (Tab. 7), MHLA further demonstrates clear advantages over other recurrent models, particularly on Multi-Doc QA, summarization, and code tasks, achieving the best average score and highlighting its strong long-context understanding capability.

Table 6: MHLA in NLP. We report results evaluated on models trained with 10B tokens. We highlight the **best** and second best entries.

| Model | CSR avg. ↑ | Wino. acc ↑ | PIQA acc ↑ | ARC-c acc_n ↑ | OBQA acc_n ↑ | ARC-e acc_n ↑ | BoolQ acc ↑ | Wiki. ppl ↓ | LMB. ppl ↓ |
|---|---|---|---|---|---|---|---|---|---|
| GLA (340M) | 46.0 | 50.0 | 62.9 | 25.5 | 31.0 | 45.8 | 60.8 | 41.47 | 86.98 |
| Transformer++ (340M) | 46.8 | 49.6 | 64.4 | 25.7 | 32.8 | 48.1 | 60.5 | **34.57** | 60.46 |
| Mamba (390M) | 46.4 | 50.5 | 64.1 | 24.9 | 32.4 | 48.3 | 58.2 | 38.32 | 62.43 |
| Mamba2 (340M) | 47.0 | 49.8 | **64.6** | 25.5 | 32.0 | **49.2** | 61.2 | 35.40 | **58.51** |
| GDN (360M) | 46.9 | **51.3** | 64.5 | 25.4 | 31.4 | 47.3 | **62.0** | 35.01 | 60.16 |
| MHLA (340M) | **47.1** | **51.3** | 64.4 | **25.9** | **33.4** | 46.5 | 61.3 | 38.31 | 71.64 |

Table 7: MHLA on LongBench. We report results evaluated on 340M models trained with 10B tokens. We highlight the **best** and second best entries

| Model | Multi-Doc QA | | | Single-Doc QA | | Few-shot | | Synthetic | | Summarization | | | Code | | AVG |
|---|---|---|---|---|---|---|---|---|---|---|---|---|---|---|---|
| | 2WM | HQA | Mus | QQA | NQA | SSM | TQA | PEN | PZH | QMS | GvR | MNs | RBP | LCC | |
| Mamba(360M) | 3.37 | 2.36 | 1.60 | 4.57 | 2.28 | 5.16 | 5.49 | 1.10 | 0.10 | 12.23 | 18.36 | 14.96 | **13.63** | 12.33 | 6.97 |
| GLA(325M) | 3.23 | 2.31 | 1.67 | 4.53 | 2.13 | 3.94 | 0.70 | **1.98** | 0.27 | 11.42 | 17.72 | 15.34 | 13.59 | 12.55 | 6.53 |
| GDN(346M) | 2.86 | 2.24 | 1.54 | **4.73** | **2.48** | **6.85** | **7.61** | 0.53 | 0.41 | 12.46 | 17.91 | **15.98** | 10.42 | 9.98 | 6.86 |
| Transformer++(325M) | **4.97** | 2.13 | **2.22** | 4.45 | 2.35 | 6.24 | 7.47 | 0.76 | 1.18 | 11.75 | 16.81 | 15.11 | 11.56 | 9.92 | 6.92 |
| Mamba2(330M) | 3.56 | 2.38 | 1.69 | 4.70 | 2.20 | 4.97 | 7.03 | 0.72 | **1.51** | 12.57 | 17.65 | 14.00 | 10.15 | 9.49 | 6.62 |
| MHLA(325M) | 3.58 | **2.97** | 1.87 | 4.68 | 2.38 | 6.41 | 6.44 | 1.69 | 1.49 | **12.58** | **18.59** | 15.01 | 13.37 | **12.72** | **7.41** |

## 5.5 ABLATION STUDY

**Multi-Head Mixing.** To evaluate the impact of our initialization strategy and learnable design in Multi-Head Mixing, we consider two variants: (1) uniform initialization without locality bias and (2) locality-biased initialization with frozen coefficients. We train and evaluate these variants on DeiT-T, with results shown in Tab. 8a. The results show that our locality-biased initialization provides a strong prior, achieving competitive performance even without learning. Allowing the coefficients to be learnable further adapts them to the dataset distribution, yielding additional performance gains.

**Head number.** We also analyze the choice of head number $M$. For DiT-S/2 at 512 resolution, the input sequence length is 1024. As discussed in Sec. 4.3, MHLA retains linear complexity when $M \leq \sqrt{1024} = 32$. We evaluate $M \in \{4, 16, 64\}$, with results summarized in Tab. 8b. MHLA achieves excellent FID already at M=16 while maintaining the highest throughput, implying that MHLA can reach best performance with a relatively small $M$ and thus leading to almost no overhead.

Table 8: **Ablation study of the proposed MHLA.**

(a) Ablation of init strategy on DeiT-T. LB-init denotes Locality-biased Initialization.

| LB-init | Learnable | Top1-acc(%) |
|---|---|---|
| | ✓ | 75.4 |
| ✓ | | 75.1 |
| ✓ | ✓ | **75.8** |

(b) Token-level head number ablation on DiT-S/2, 512px.

| Head number | FID↓ | Throughput↑ |
|---|---|---|
| 4 | 79.56 | 435 |
| 16 | **78.63** | **435** |
| 64 | 79.50 | 408 |

## 6 CONCLUSION

In this paper, we introduce a novel linear attention mechanism, termed **Multi-Head Linear Attention (MHLA)**. By partitioning tokens into multiple groups, MHLA effectively preserves token-wise diversity. Without relying on additional modules such as depthwise convolutions or hybrid self-attention layers, MHLA achieves performance comparable to or even surpassing that of self-attention-based models. We envision this work as establishing a fundamental attention mechanism that can benefit a wide range of downstream applications, such as high-quality image generation, long-horizon video synthesis, and large-scale language modeling.

**Future improvement.** Although MHLA demonstrates strong performance on image classification and AIGC, our current NLP experiments are limited to a 0.3B model and mainly serve as a preliminary validation. A key future direction is to evaluate MHLA at larger language model scales to better understand its scaling behavior in realistic LLM settings.

## ETHICS STATEMENT

This paper does not involve studies with human subjects, and it does not raise any concerns regarding harmful insights, discrimination, or privacy issues. The methods employed focus on improving the efficiency of transformer models in machine learning tasks such as image classification, generation, and natural language processing. No conflicts of interest are present, and the research adheres to the highest standards of scientific integrity.

## REPRODUCIBILITY STATEMENT

To ensure reproducibility, the authors have provided sufficient details on the methods and experimental setup. The MHLA implementation and experiments are described in detail in the paper. The code will be publicly available once the paper is accepted. All experiments, including image classification and generation tasks, are reproducible as they adhere to standard benchmarks (e.g., ImageNet-1K) and configurations from previous work. The authors have also included the results for different model architectures and configurations, demonstrating consistency across various tasks. Further, the supplementary appendix provides additional implementation details to facilitate reproduction.

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

## A    FULL RELATED WORKS

**Transformer.**    Since the introduction of the Transformer architecture (Vaswani et al., 2017), self-attention has become the dominant mechanism across a wide range of domains, including natural language processing (Devlin et al., 2019; Brown et al., 2020), computer vision (Dosovitskiy et al., 2021; Liu et al., 2021; Hou et al., 2021; Zhou et al., 2021), and generative modeling (Esser et al., 2021; Saharia et al., 2022). The expressive power of self-attention stems from its ability to model pairwise interactions among all tokens, but this comes at a quadratic cost in both computation and memory. This limitation becomes particularly pronounced in large-scale or real-time applications, motivating the exploration of more efficient attention mechanisms. A broad spectrum of strategies has been proposed, such as sparse attention (Child et al., 2019; Beltagy et al., 2020; Zaheer et al., 2020), low-rank approximations (Wang et al., 2020; Xiong et al., 2021), and hardware-optimized variants such as FlashAttention (Dao et al., 2022; Dao, 2024). Despite these advances, designing efficient attention mechanisms that maintain both scalability and accuracy remains an open challenge.

**Linear Attention.**    Linear attention has emerged as a prominent direction for addressing the quadratic complexity of standard self-attention. Early works reformulated the softmax operation with kernel-based feature mappings, enabling linear-time complexity in both training and inference (Katharopoulos et al., 2020; Choromanski et al., 2021; Peng et al., 2023; 2024; Yang et al., 2024). While these approaches make Transformers scalable to long sequences, they often suffer from reduced representational power compared to full softmax attention, leading to accuracy drops in challenging tasks such as vision and generative modeling. To bridge this gap, subsequent research has incorporated additional modules to enrich the expressiveness of linear attention. For example, convolutional layers have been introduced to capture local context (Peng et al., 2021; Shen et al., 2021; Han et al., 2023; Fan et al., 2025b), gating mechanisms have been proposed to better control information flow. More recently, state space models such as Mamba (Gu & Dao, 2023; Dao & Gu, 2024) and its variants (Shi et al., 2024; Liu et al., 2024) have also been explored as efficient alternatives to linear attention, showing strong scalability on long sequences and competitive accuracy. However, these methods still face two fundamental limitations: (1) when applied in a unidirectional form to tasks requiring bidirectional attention, they exhibit substantial performance degradation; and (2) when augmented with extra modules (e.g., convolutional layers or additional self-attention blocks), they inevitably incur higher computational overhead and remain vulnerable to *global context collapse* (see Sec. 3.2), where the global summary loses representational diversity

**Sparse Attention.**    In addition to linear attention, sparse attention mechanisms have been another major approach to addressing the computational bottleneck in Transformers. Methods such as Longformer (Beltagy et al., 2020) and BigBird (Zaheer et al., 2020) introduce sparse attention patterns, where each token only attends to a subset of the other tokens, reducing the overall number of attention operations. These methods exploit structural sparsity (e.g., local or global attention patterns) to maintain efficiency while still capturing global context in long sequences. Other techniques, such as the Performer (Choromanski et al., 2021), propose using kernel approximations to achieve sparse attention while preserving the model's expressive power. Although sparse attention mechanisms improve scalability, they often introduce trade-offs in terms of accuracy, especially in tasks requiring full token interactions.

**Applications of Linear and Sparse Attention.**    Linear and sparse attention mechanisms have been successfully applied across various domains, including NLP, CV, and generative modeling. In NLP, linear attention has been used to scale models like BERT (Devlin et al., 2018) and GPT (Radford et al., 2019) to longer sequences, enabling better handling of long documents and improving efficiency in language models (Devlin et al., 2019; Brown et al., 2020). In computer vision, linear attention methods have been applied to vision transformers to improve efficiency when processing large images, as seen in works like Swin Transformer (Liu et al., 2021) and DeiT (Touvron et al., 2021). These applications demonstrate the broad utility of linear and sparse attention mechanisms, but also highlight the need for continued development to balance efficiency with the expressive power required by complex tasks like image generation and video understanding.

## B  QUERY-CONDITIONED SELECTIVITY IN SOFTMAX ATTENTION

A key advantage of softmax self-attention is its *query-conditioned selectivity*. Recall the standard attention formulation:

$$\text{Attn}(Q, K, V)_i = \sum_{j=1}^{N} \alpha_{ij} v_j, \qquad \alpha_{ij} = \frac{\exp(q_i^\top k_j)}{\sum_{t=1}^{N} \exp(q_i^\top k_t)}.$$

Two properties are crucial: (i) **Query-conditioned weighting:** each query $q_i$ produces its own distribution $\{\alpha_{ij}\}_{j=1}^{N}$, so the relative importance of token $k_j$ is fully dependent on $q_i$; (ii) **Per-token weighting:** the weights act directly on each $v_j$, without collapsing $V$ into a global summary. Together, these properties give softmax attention the ability to produce highly adaptive, sharply concentrated context vectors.

By contrast, *global linear attention* aggregates all tokens into a single summary matrix $S^{\text{global}} = \sum_{j=1}^{N} \widetilde{K}_j V_j^\top$ shared by all queries, yielding

$$\text{Attn}_{\text{lin}}(Q, K, V)_i = \frac{\widetilde{q}_i^\top S^{\text{global}}}{\widetilde{q}_i^\top \left(\sum_{j=1}^{N} \widetilde{K}_j\right)},$$

where the per-token contributions are no longer explicitly separable by $i$. As a result, different queries obtain nearly identical context vectors, losing query-conditioned selectivity.

**MHLA restores query-conditioned selectivity.**  MHLA bridges this gap by introducing a learnable coefficient matrix $\mathcal{M}_c$ that forms *query-block-specific mixtures* of local summaries:

$$\widetilde{S}_i = \sum_{b=1}^{M} m_{i,b} S_b \qquad \Rightarrow \qquad \text{Attn}_{\text{MHLA}}(Q, K, V)_i = \widetilde{q}_i^\top \widetilde{S}_i.$$

Because $m_{i,b}$ varies with the query block $i$, MHLA assigns different effective weights to the same token depending on the querying block. Expanding $S_b$ into its token-level definition gives

$$\widetilde{q}_i^\top \widetilde{S}_i = \sum_{t=1}^{N} m_{i,b(t)} \left(\widetilde{q}_i^\top \widetilde{K}_t\right) V_t^\top,$$

revealing a two-stage weighting mechanism: (i) block-level selection $m_{i,b(t)}$ that is query-conditioned, followed by (ii) within-block token reweighting via the kernel inner product $\widetilde{q}_i^\top \widetilde{K}_t$. This design reintroduces query-conditioned selectivity and per-token weighting while preserving the linear-time complexity of kernelized attention.

## C  MHLA FOR AUTOREGRESSIVE MODELING

In autoregressive modeling, the causal mask prevents each token from attending to future tokens. While linear attention normally achieves $O(Nd^2)$ complexity by reusing a global key–value summary, under causal masking, the summary must be recomputed or updated for each prefix, which naively results in $O(N^2 d)$ cost over the full sequence. To avoid this quadratic overhead, a widely adopted solution for linear attention is *chunkwise parallel training* (Sun et al., 2023), which splits the sequence into blocks of size $C$ and processes them in parallel to avoid the quadratic cost of recomputing attention over all past tokens. For block $b$, a local key–value summary is computed as $S_b = \sum_{j \in b} \widetilde{K}_j V_j^\top \in \mathbb{R}^{d \times d}$, and the global summary is updated recursively:

$$S_i^{\text{global}} = S_{i-1}^{\text{global}} + S_i, \qquad H_i = Q_i S_{i-1}^{\text{global}} + (Q_i \widetilde{K}_i^\top) V_i.$$

Here, the first term propagates context from preceding blocks via the prefix summary $S_{i-1}^{\text{global}}$, while the second term captures intra-block attention. This chunkwise scheme preserves causality and allows block-parallel training with per-block complexity $O(Cd^2 + C^2 d)$, leading to an overall cost $O\left(\frac{L}{C}(Cd^2 + C^2 d)\right)$ for a sequence of length $L$.

**MHLA with chunkwise parallel training.** MHLA extends this scheme by replacing the single global summary with *query-conditioned mixtures* of local summaries. Specifically, for block $i$ we form a mixed summary

$$\widetilde{S}_i = \sum_{b \leq i} m_{i,b} S_b, \qquad H_i = Q_i \widetilde{S}_{i-1} + m_{i,b}(Q_i \widetilde{K}_i^\top) V_i.$$

where $m_{i,b}$ are the learnable mixing coefficients from the causal coefficient matrix $\mathcal{M}_c^{\text{causal}}$ (upper-triangular entries masked to enforce causality). Queries in block $i$ then interact only with $\widetilde{S}_i$, yielding block-specific, query-adaptive context representations rather than a shared global one. Because the mixing is performed once per block and reused for all tokens in that block, the asymptotic complexity matches that of chunkwise linear attention.

**Causal inference.** At inference time, we maintain the set of past local summaries $\{S_1, \ldots, S_{i-1}\}$ and incrementally update the current block summary $S_i$ as new tokens arrive. When a block is complete, its contribution to future mixtures is fixed and cached. For a new token in block $i$, we simply update $S_i \leftarrow S_i + \widetilde{K}_t V_t^\top$ and recompute the block's mixed summary $\widetilde{S}_i$ by applying $m_{i,i}$ to the incremental update. This avoids recomputation over previous blocks and keeps per-token complexity $O(d^2)$.

# D  DATASET

To assess the effectiveness of our approach, we conduct extensive experiments on four tasks: image classification, class-to-image (C2I) generation, text-to-image (T2I) generation, and natural language processing. Following prior works (Fan et al., 2025a;b; Han et al., 2023), we train classification and C2I models on ImageNet-1K (Deng et al., 2009) and evaluate them on the standard validation set. For T2I generation, we finetune a pretrained model using a relative small collection of 31,292k images gotten from the internet. For natural language processing, we train models with a subset of SlimPajama (Shen et al., 2024) with 5B tokens.

# E  EXTRA IMPLEMENTATION DETAILS

**Image Classification.** For training of DeiT, we replace the class token with average pooling and train all baselines under identical settings to ensure fair comparison. We additionally add CPE (Chu et al., 2021) with a kernel size of 3, following previous works for a fair comparison. For VLT, we strictly follow the setup in (Fan et al., 2025b). All models are trained for 300 epochs with a batch size of 1024 and a peak learning rate of 1e-3. For models with an input size of 224, we pad the input size to 256 for better splitting of heads. The head number $M$ is set to 16 for DeiT modes. For VLT models, the sequence length for the two linear attention layers is $\{3136, 784\}$. So we set the head number $M$ to $\{49, 16\}$ for the two layers respectively.

# F  COMPLETE EXPERIMENTAL RESULTS

## F.1  IMAGE GENERATION

We illustrate the complete results on DiT and DiG models in Tab. 10 and Tab. 9. We provide more generation results of SANA-MHLA in Fig. 7.

Table 9: **Fast adaptation results on DiT-XL/2 with MHLA, with and without guidance.**

| Model | Attention Type | Resolution | FID ↓ | IS ↑ | sFID ↓ | Precision ↑ | Recall ↑ |
|---|---|---|---|---|---|---|---|
| DiT-XL/2 | Self Attention | 256 | 9.62 | 121.50 | 6.85 | 0.67 | 0.67 |
| | MHLA (Ours) | 256 | 8.34 | 121.27 | 5.52 | 0.69 | 0.65 |
| DiT-XL/2(G) | Self Attention | 256 | 2.27 | 278.24 | 4.60 | 0.83 | 0.57 |
| | MHLA (Ours) | 256 | 2.54 | 252.07 | 4.67 | 0.83 | 0.56 |

Table 10: **Comparison of different attention types across models.**

| Model | Attention Type | Resolution | FID ↓ | IS ↑ | sFID ↓ | Precision ↑ | Recall ↑ |
|-------|----------------|------------|-------|------|--------|-------------|----------|
| DiT-S/2 | Self Attention | 256 | 68.40 | – | – | – | – |
| | Linear Attention | 256 | 89.72 | 15.24 | 21.87 | 0.28 | 0.41 |
| | MHLA (Ours) | 256 | **59.80** | 23.49 | 10.16 | 0.39 | 0.56 |
| | Self Attention | 512 | 84.54 | 15.53 | 17.02 | 0.36 | 0.49 |
| | Linear Attention | 512 | 125.33 | 33.11 | 11.64 | 0.22 | 0.29 |
| | MHLA (Ours) | 512 | **78.63** | 13.11 | 18.50 | 0.40 | 0.49 |
| DiG-S/2 | GLA (Yang et al., 2024) | 256 | 62.06 | – | – | – | – |
| | GLA | 512 | 99.04 | – | – | – | – |
| | MHLA (Ours) | 256 | **59.49** | 24.04 | 11.51 | 0.40 | 0.57 |
| DiT-B/2 | Self Attention | 256 | 43.47 | – | – | – | – |
| | Linear Attention | 256 | 60.47 | 24.27 | 13.69 | 0.39 | 0.57 |
| | MHLA (Ours) | 256 | **37.47** | 38.79 | 7.35 | 0.51 | 0.63 |
| DiT-L/2 | Self Attention | 256 | 23.33 | – | – | – | – |
| | Linear Attention | 256 | 32.35 | 45.57 | 8.55 | 0.54 | 0.62 |
| | MHLA (Ours, w/None) | 256 | 25.37 | 54.38 | 6.06 | 0.59 | 0.61 |
| | MHLA (Ours, w/ CPE) | 256 | 24.21 | 57.62 | 6.12 | 0.59 | 0.62 |
| | MHLA (Ours, w/ CPE+Gating) | 256 | **21.37** | 63.47 | 5.80 | 0.61 | 0.62 |
| DiT-XL/2 | Self Attention | 256 | 19.47 | – | – | – | – |
| | Linear Attention | 256 | 28.63 | 51.15 | 8.23 | 0.57 | 0.62 |
| | MHLA (Ours, w/ None) | 256 | 20.32 | 65.95 | 6.01 | 0.61 | 0.62 |
| | MHLA (Ours, w/ CPE) | 256 | 22.79 | 61.80 | 5.53 | 0.60 | 0.62 |
| | MHLA (Ours, w/ CPE+Gating) | 256 | **19.17** | 68.97 | 5.70 | 0.63 | 0.62 |

Table 11: Comparison with LiT. We report the FID scores (mean ± std) over three independent runs for MHLA to demonstrate result stability.

| Model | FID (mean ± std) |
|-------|------------------|
| LiT-S/2 | 63.21 |
| DiT-S/2 w/ MHLA | $59.74 \pm 0.10$ |
| LiT-B/2 | 40.86 |
| DiT-B/2 w/ MHLA | $37.52 \pm 0.04$ |
| LiT-L/2 | 24.04 |
| DiT-L/2 w/ MHLA | $21.43 \pm 0.05$ |
| LiT-XL/2 | 20.66 |
| DiT-XL/2 w/ MHLA | $19.16 \pm 0.03$ |

We additionally provide more comprehensive comparisons against other recent linear attention methods on image generation tasks (Wang et al., 2025), and report the mean and standard deviation of MHLA over three independent runs to demonstrate the stability of our results. The corresponding results are summarized in Tab. 11.

## F.2 ABLATION OF CPE AND OUTPUT GATING.

Table 12: Ablation study of MHLA with CPE and output gating.

| Setting | FID |
|---------|-----|
| Linear Attention | 89.7 |
| MHLA w/ None | 76.4 |
| MHLA w/ CPE | 64.0 |
| MHLA w/ Gating | 68.5 |
| MHLA w/ CPE+Gating | **59.8** |

We conducted a detailed analysis of the effects of CPE and Output Gating when combined with MHLA in the DiT-S model, as shown in Tab. 12. Our findings show that, in smaller models, CPE and Output Gating serve as orthogonal optimizations of MHLA, effectively enhancing the expressive

Table 14: Training throughput (tokens/s) on H100 under different sequence length $\times$ batch size configurations.

| Type | Model | 2K$\times$16 | 4K$\times$8 | 8K$\times$4 | 16K$\times$2 |
|------|-------|------|-----|-----|------|
| | MHLA | 83249 | 83677 | 81762 | 81427 |
| PyTorch w/o optimized kernel | GDN | 28763 | 19217 | 12306 | 7262 |
| | GLA | 57815 | 31875 | 18092 | 17375 |
| | GDN | 81437 (2.8$\times$) | 81596 | 80739 | 78049 |
| PyTorch w/ optimized kernel | GLA | 141386 (2.5$\times$) | 139735 | 139871 | 137635 |
| | Mamba2 | 96871 | 96787 | 96911 | 96450 |

ability when the model size is insufficient. However, our experiments in Tab. 3a indicate that the performance gains from CPE and Output Gating diminish as the model size increases. In the DiT-XL model, adding CPE alone actually leads to a performance decrease. In contrast, MHLA consistently provides significant improvements in expressivity, regardless of model size.

### F.3    CLASSIFICATION RESULTS ON HIGHER RESOLUTIONS

We further conducted additional experiments at resolutions of 384$\times$384 and 512$\times$512, using the DeiT-T model to verify the effectiveness of MHLA on high-resolution classification tasks. Results are shown in Tab. 13.

Table 13: High-resolution classification accuracy of DeiT-T with and without MHLA.

### F.4    SCALING ANAYLSIS

| Model | Resolution | ACC |
|-------|-----------|-----|
| DeiT-T | 384$\times$384 | 74.4 |
| DeiT-T + MHLA | 384$\times$384 | **77.5** |
| DeiT-T | 512$\times$512 | 75.3 |
| DeiT-T + MHLA | 512$\times$512 | **78.3** |

In this section, we conduct empirical studies to evaluate the throughput of MHLA across different tasks under varying sequence lengths N and token-level head numbers M. The results in Tab. 16 and Tab. 17 show that when $M^2 < N$ is satisfied, MHLA introduces only negligible overhead, whereas larger M leads to more noticeable overhead. However, our ablation studies in Tab. 8b have already demonstrated that choosing M such that $M^2 < N$ is sufficient to achieve strong performance.

### F.5    STREAMING AND CAUSAL INFERENCE EFFICIENCY

To empirically validate MHLA's compatibility with chunkwise-parallel training and streaming/stateful execution under causal masking, we benchmark both training throughput and autoregressive inference latency on NVIDIA H100 GPUs.

**Training Throughput.**    We report tokens per second (tokens/s) under different "sequence length $\times$ batch size" configurations in Table 14.  Under the standard PyTorch implementation without optimized kernels, MHLA consistently achieves the highest throughput across all settings (2K$\times$16 to 16K$\times$2), significantly outperforming GLA and GDN.

Although we have not yet implemented fused or customized CUDA kernels for MHLA due to time constraints, prior results on GLA and GDN demonstrate that kernel-level optimization can provide approximately 2.5$\times$–3$\times$ acceleration. Since MHLA is built upon the same GEMM-based primitives, comparable speedup is expected after similar optimization.

**Inference Latency.**    To further evaluate streaming inference efficiency, we measure causal autoregressive latency at sequence length 4096. As shown in Table 15, under the unoptimized PyTorch implementation, MHLA achieves the lowest latency (204.3s), outperforming both GLA and GDN, and even surpassing the heavily optimized Mamba2 baseline.

These results demonstrate that MHLA not only maintains theoretical compatibility with chunkwise-parallel training and streaming execution, but also delivers strong empirical efficiency under causal masking.

Table 15: Autoregressive inference latency (seconds) on H100 with sequence length 4096. MHLA uses head number $M = 32$.

| Type | Model | Latency (s) |
|------|-------|-------------|
| PyTorch w/o optimized kernel | MHLA | 204.3 |
| | GLA | 269.4 |
| | GDN | 292.3 |
| PyTorch w/ optimized kernel | GDN | 126.8 |
| | GLA | 144.4 |
| | Mamba2 | 311.0 |

Table 16: Profiling results of MHLA under varying sequence length $N$ and token-level head number $M$. Left: DiT-S/2. Right: DeiT-S/16.

| M\N | 256 | 1024 | 4096 |
|-----|-----|------|------|
| 4 | 42ms 3.7G | 52ms 7.1G | 147ms 20.8G |
| 16 | 40ms 3.9G | 51ms 7.2G | 145ms 21.0G |
| 64 | 39ms 4.8G | 52ms 8.0G | 148ms 21.7G |
| 256 | – | 61ms 12.0G | 157ms 25.4G |
| 1024 | – | – | 219ms 40.0G |

| M\N | 256 | 1024 |
|-----|-----|------|
| 4 | 129 imgs/s 3.4G | 124 imgs/s 8.9G |
| 16 | 118 imgs/s 3.8G | 118 imgs/s 9.4G |
| 64 | 150 imgs/s 5.7G | 104 imgs/s 11.0G |
| 256 | – | 89 imgs/s 18.0G |

# G    CLARIFICATION ON TERMINOLOGY AND COMPUTATIONAL CONCEPTS

In this section, we provide formal definitions for the terminology used in our method. These terms describe novel computational behaviors in MHLA that lack direct analogues in prior linear attention formulations.

## G.1    CONCEPT 1: *query-conditioned*

The phrase "query-conditioned" describes a mechanism where the aggregation of contextual information is dynamic and specific to each query instance, distinct from the fixed recurrence found in standard linear attention.

Specifically, the process operates as follows:

- Each query token is associated with a unique vector of mixing coefficients.

- These coefficients are used to weight and aggregate all local KV summaries independently for every query position.

Consequently, the adaptation occurs *per query*, rather than globally or via a shared recursive rule.

## G.2    CONCEPT 2: *KV Summary vs. Hidden States*

We introduce the term KV Summary tos strictly distinguish our approach from the Hidden State found in traditional linear attention papers. While the KV summary may seemingly resemble Hidden States in notation, the underlying computation and dependency graphs are structurally different in two key aspects:

- Unlike the strict recursive chain in traditional linear attention where $h_t$ relies on $h_{t-1}$, MHLA computes each Global KV Summary ($S_g$) independently, eliminating state propagation across positions.

- While traditional states are derived via a one-to-one update from the previous step, MHLA follows a many-to-one aggregation pattern, where each $S_g$ is computed from *all* local summaries using specific mixing coefficients.

By avoiding the rigid inheritance of history inherent to hidden states, MHLA's KV summaries achieve greater expressivity and flexibility.

Table 17: Scaling analysis results for the language model. We report the theoretically derived values, aligned with empirical measurements. We believe that further engineering optimizations will follow the same trend and achieve stronger performance.

| M/N | 2048 | 4096 | 8192 |
|-----|------|------|------|
| 16 | 11100 tokens/s 4.83G | 22000 tokens/s 7.46G | 40500 tokens/s 13.04G |
| 32 | 11000 tokens/s 4.99G | 21800 tokens/s 7.55G | 39700 tokens/s 12.84G |
| 64 | – | 21100 tokens/s 7.89G | 39400 tokens/s 13.06G |
| 128 | – | – | 29600 tokens/s 13.79G |

Figure 7: **More generation results from our fine-tuned SANA-MHLA model.**

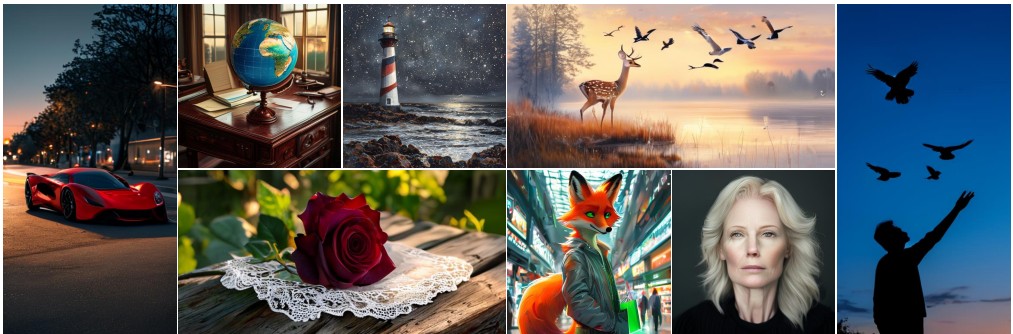

# H    LLM USAGE.

We used large language models (LLMs) solely as a writing aid to polish the clarity and readability of the manuscript. Specifically, we employed LLM-based tools to (i) refine grammar and phrasing for academic style consistency, (ii) improve logical flow between sections, and (iii) condense overly verbose passages. No new research ideas, experimental designs, or results were produced by the LLM; all scientific contributions, methodology development, and experimental analyses were conceived and executed by the authors.

