# OpenReview forum: "MHLA: Restoring Expressivity of Linear Attention via Token-Level Multi-Head"
_ICLR.cc/2026/Conference — ICLR 2026 Poster_

### Official Review · Reviewer_2bkY · 2025-10-30

**Soundness:** 3
**Presentation:** 3
**Contribution:** 2
**Rating:** 4
**Confidence:** 2

**Summary:**

This paper addresses a key limitation in linear attention models, termed global context collapse, where the model’s representational diversity diminishes as sequence length increases.
To overcome this, the authors propose Multi-Head Linear Attention (MHLA), which partitions the token sequence into multiple blocks and computes local key–value summaries within each.
By introducing a multi-head mixing mechanism that adaptively combines these local summaries, MHLA restores query-conditioned diversity while maintaining linear computational complexity.
Extensive experiments demonstrate that MHLA achieves significant improvements over existing linear attention baselines in image classification (ImageNet-1K), image generation (DiT and SANA models), and NLP benchmarks, with minimal computational overhead.

**Strengths:**

1. Insightful analysis of linear attention – The authors conduct a rigorous examination of why linear attention underperforms, identifying global context collapse as the root cause. Their analysis of rank deficiency and entropy in attention maps provides a strong theoretical foundation.

2. Innovative multi-head design – MHLA introduces a token-level multi-head mechanism that mixes local key–value summaries through learnable coefficients. This design cleverly restores query-dependent diversity and sparsity while keeping linear complexity.

3. Strong empirical results – On ImageNet classification, MHLA improves accuracy by 3.6% over standard linear attention, achieving state-of-the-art results when integrated into DeiT and VLT architectures. On image generation, it delivers up to 12.6% gains over baselines while matching or surpassing softmax attention performance in large-scale DiT models. On NLP tasks, MHLA yields 2.1% improvements over existing linear attention methods, even without additional positional embeddings.

**Weaknesses:**

### 1. Limited and Inconclusive Experimental Evaluation
The major limitation of this paper lies in its lack of comprehensive experiments to support its central claim—improving long-term modeling capability.
In the vision domain, experiments are restricted to image classification on 224×224 inputs, which are too short and fail to reflect long-range dependencies.
In the NLP domain, the evaluation covers only three small-scale reasoning benchmarks (ARC-c, WinoGrande, CoPA), which do not assess long-context understanding, open-ended text generation, or large-scale language modeling.
Consequently, it remains unclear whether MHLA truly benefits long-sequence scaling.
Moreover, the NLP model used (340 M parameters trained on 5 B tokens) is relatively small; while suitable for controlled experiments, it is insufficient to demonstrate MHLA’s effectiveness in large-scale language models, where long-context efficiency becomes critical.
To better validate the method’s claims, I recommend incorporating:

Vision tasks such as image captioning, VQA, or video captioning, which require multi-modal and temporal reasoning.

NLP datasets such as LongBench, PG19, or BookSum, which directly evaluate scalability and diversity preservation in long-sequence scenarios.

### 2. Incomplete Discussion of the Computational Trade-off
The proposed method essentially trades extra computation for improved representational rank.
This trade-off is reasonable when the additional $O(M^2d^2)$ cost remains small, but it does not fundamentally solve the scalability problem for very long input sequences. Linear attention mechanisms are expected to handle arbitrarily long contexts, yet MHLA’s computational complexity scales with block size $M$, limiting its practicality for extremely long sequences.
The paper should more clearly analyze this scaling limitation and discuss how MHLA behaves as sequence length increases.

### 3. No Throughput or Speed Analysis
The paper does not report throughput, latency, or efficiency comparisons.
Although MHLA maintains linear complexity in theory, the method introduces an additional $O(M^2d^2)$ term due to multi-head block mixing.
This overhead may be negligible in small-scale settings like image classification or ARC-c, but it could become significant in truly long-sequence scenarios.
Without explicit runtime or memory benchmarks, the paper fails to demonstrate that MHLA preserves an effective balance between accuracy and computational cost under long-context conditions.

### 4. Weak Literature Review and Terminology Issues
The paper’s literature review is notably shallow.
The introduction and related work sections lack a comprehensive discussion of the extensive prior research on linear attention, a highly active field with numerous variants and analyses.
The authors should consult and reference existing survey papers on linear attention to position MHLA more accurately within the broader landscape.
Additionally, the writing style introduces several non-standard or self-coined terms—such as “adapt to each query individually” and “KV summary”—which are not widely adopted in the community.
The authors are encouraged to revise these descriptions using standard terminology and provide clearer, community-recognized definitions to improve readability and academic rigor.

**Questions:**

See weakness

---

> ### Author Response · Authors · 2025-11-23
> **Response to Reviewer 2bkY (1/4)**
>
> **W1: Limited and Inconclusive Experimental Evaluation**
>
> > In the vision domain, experiments are restricted to image classification on 224×224 inputs, which are too short and fail to reflect long-range dependencies. In the NLP domain, the evaluation covers only three small-scale reasoning benchmarks, which do not assess long-context understanding, open-ended text generation, or large-scale language modeling. Consequently, it remains unclear whether MHLA truly benefits long-sequence scaling. Moreover, the NLP model used is relatively small; while suitable for controlled experiments, it is insufficient to demonstrate MHLA’s effectiveness in large-scale language models, where long-context efficiency becomes critical.
> >
>
> Thank you for the insightful comments regarding both the vision and NLP experiments. We have added experiments across image classification, video generation and language modeling tasks to solve your concern.
>
> 1. **Extended High-Resolution Classification Experiments**
>
> Common image classification benchmarks evaluate efficient attention methods on ImageNet at 224x224 resolution, which is chosen by us to ensure fair comparison. To further address your concern, we additionally include classification results at 384x384 and even higher 512x512, testing both the baseline and MHLA under these more demanding settings. MHLA consistently achieves higher accuracy and better efficiency across all resolutions.
>
> | Model | Resolution | ACC$\uparrow$ |
> | --- | --- | --- |
> | Deit-T | 384x384 | 74.4 |
> | DeiT-T with MHLA | 384x384 | 77.5 |
> | Deit-T | 512x512 | 75.3 |
> | DeiT-T with MHLA | 512x512 | 78.3 |
>
> 2. **Additional Long-Context Vision Task**
>
> Moreover, among vision tasks, video generation critically requires extremely long sequence modeling. We therefore include additional results on video generation using MHLA. We finetune  Wan2.1-1.3B by changing  Flash Attention to MHLA and evaluated it on VBench (**context length up to 32k**). Result presented in the table below further verifies the strong long-context modeling ability of MHLA. MHLA notably **achieves 2.1x speedup** than the flash-attention based method, keeping the same latency with vanilla linear attention, while also attaining comparable *overall* and *semantic VBench scores*, demonstrating its strong expressivity on long context.
>
> | Method | Quality score$\uparrow$ | Semantic score$\uparrow$ | Total$\uparrow$ | Latency$\downarrow$ |
> | --- | --- | --- | --- | --- |
> | Wan2.1 1.3B | 85.23 | 75.65 | 83.31 | 166s |
> | Full MHLA | 84.26 | 76.16 | 82.62 | 81s |
> | Full Linear  | 69.96 | 11.38 | 58.24 | 82s |
> | MHLA Hybrid 2/3 | 84.87 | 79.59 | 83.82 | 103s |
>
> *Note: Full MHLA and Full Linear indicate changing all flash attention to MHLA / vanilla linear attention. MHLA Hybrid 2/3 means changing 2/3 layer to MHLA. Both MHLA and vanilla linear attention are accelerated with torch.compile.*
>
> 3. **Additional NLP Long-Context Evaluation**
>
> For the NLP task, we increase the training token budget and train the 340M model on 10B tokens. To verify the long-sequence modeling ability of MHLA, we evaluate it on LongBench and compare it against more recent SOTA architectures such as Gated DeltaNet (GDN) [1] and Mamba2 [2], achieving the best average score, showing a comparable performance with SOTAs, as presented in the table below.
>
> |  | Overall Performance | Multi-Doc QA |  |  | Summarization |  |  | Code |  | Single-Doc QA |  | Few-shot |  | Synthetic Task |  |
> | --- | --- | --- | --- | --- | --- | --- | --- | --- | --- | --- | --- | --- | --- | --- | --- |
> | Model | avg $\uparrow$ | hotpotqa $\uparrow$ | musique $\uparrow$ | 2wikimqa $\uparrow$ | qmsum $\uparrow$ | gov_report $\uparrow$ | multi_news $\uparrow$ | repobench-p $\uparrow$ | lcc $\uparrow$ | qasper $\uparrow$ | nqa $\uparrow$ | samsum $\uparrow$ | triviqa $\uparrow$ | passage retrieval en $\uparrow$ | passage retrieval zh $\uparrow$ |
> | Mamba | *6.97* | 2.36 | 1.60 | 3.37 | 12.23 | *18.36* | 14.96 | **13.63** | 12.33 | 4.57 | 2.28 | 5.16 | 5.49 | 1.1 | 0.1 |
> | GLA | 6.53 | 2.31 | 1.67 | 3.23 | 11.42 | 17.72 | 15.34 | *13.59* | 12.55 | 4.53 | 2.13 | 3.94 | 0.7 | **1.98** | 0.27 |
> | GDN | 6.86 | 2.24 | 1.54 | 2.86 | 12.46 | 17.91 | **15.98** | 10.42 | 9.98 | **4.73** | **2.48** | **6.85** | **7.61** | 0.53 | 0.41 |
> | Transformer++ | 6.92 | **2.13** | **2.22** | **4.97** | 11.75 | 16.81 | *15.11* | 11.56 | 9.92 | 4.45 | 2.35 | 6.24 | *7.47* | 0.76 | 1.18 |
> | Mamba2 | 6.62 | *2.38* | 1.69 | 3.56 | *12.57* | 17.65 | 14.00 | 10.15 | 9.49 | *4.70* | 2.20 | 4.97 | 7.03 | 0.72 | **1.51** |
> | MHLA | **7.41** | **2.97** | *1.87* | *3.58* | **12.58** | **18.59** | 15.01 | 13.37 | **12.72** | 4.68 | *2.38* | *6.41* | 6.44 | *1.69* | *1.49* |

---

> ### Author Response · Authors · 2025-11-23
> **Response to Reviewer 2bkY (2/4)**
>
> We also compare MHLA with baselines with a wider range of benchmarks. The result is presented below. MHLA shows SOTA performance on Common-sense reasoning and gains the best score in MMLU.
>
>
> | Tasks | MMLU acc $\uparrow$ | **CSR .avg score** $\uparrow$ | Wino acc $\uparrow$ | PIQA acc $\uparrow$ | ARC-c acc_n $\uparrow$ | OBQA acc_n $\uparrow$ | ARC-e acc_n $\uparrow$ | BoolQA acc $\uparrow$ | Wiki ppl $\downarrow$ | LMB ppl $\downarrow$ |
> | --- | --- | --- | --- | --- | --- | --- | --- | --- | --- | --- |
> | GLA (340M) | 22.9 | 46.0 | 50.0 | 62.9 | 25.5 | 31.0 | 45.8 | 60.8 | 41.47 | 86.98 |
> | Transformer++(340M) | 22.9 | 46.8 | 49.6 | 64.4 | 25.7 | 32.8 | 48.1 | 60.5 | 34.57 | 60.46 |
> | Mamba (390M) | 23.5 | 46.4 | 50.5 | 64.1 | 24.9 | 32.4 | **48.3** | 58.2 | 38.32 | 62.43 |
> | Mamba2 (340M) | 23.0 | 47.0 | 49.8 | **64.6** | 25.5 | 32.0 | 49.2 | 61.2 | 35.40 | 58.51 |
> | GDN (360M) | 23.0 | 46.9 | **51.3** | 64.5 | 25.4 | 31.4 | 47.3 | **62.0** | 35.01 | 60.16 |
> | MHLA (340M) | **23.7** | **47.1** | **51.3** | 64.4 | **25.9** | **33.4** | 46.5 | 61.3 | 38.31 | 71.64 |
>
> [1] Gated Delta Networks: Improving Mamba2 with Delta Rule. ICLR 2025.
>
> [2] Transformers are SSMs: Generalized Models and Efficient Algorithms Through Structured State Space Duality. ICML 2024.
>
> **W2: Incomplete Discussion of the Computational Trade-off.**
>
> > Linear attention mechanisms are expected to **handle arbitrarily long contexts**, yet MHLA’s computational complexity scales with block size, limiting its practicality for extremely long sequences. The paper should more clearly analyze this scaling limitation and discuss how MHLA behaves as sequence length increases.
> >
>
> We appreciate the reviewer’s suggestion. However, we would like to highlight that MHLA can indeed handle an arbitrarily long context. We explain it in detail here. In MHLA, **block size scales with sequence length to always keep linear complexity.** Our experiment on video generation (32k sequence length) exactly verifies this point (with result in response to w1). By selecting a larger block size proportional to the sequence length, we achieved superior performance while maintaining the same efficiency as vanilla linear attention
>
> **Scalability Analysis**.  As emphasized in Section 3.3. Regarding the scalability of MHLA, its computational complexity is $O(N d^{2} + M^{2} d^{2})$, where N denotes the sequence length, m is the number of token-level heads, and the block size is defined as $\frac N M$ . In all our experiments, we explicitly enforce the constraint $M^{2} ≤ N$. By definition of time complexity, this implies $O(N d^{2} + M^{2} d^{2}) ≤ O(N d^{2} + N d^{2}) = O(N d^{2})$, meaning that under the condition $M^{2} ≤ N$, the **overall complexity is still linear with N**. We have also shown in **Table 6(b)**, through ablation studies, that choosing a head number $M \ll \sqrt{N}$ already provides sufficient representational capacity.

---

> ### Author Response · Authors · 2025-11-23
> **Response to Reviewer 2bkY (3/4)**
>
> **W3: No Throughput or Speed Analysis.**
>
> > The paper does not report throughput, latency, or efficiency comparisons. Although MHLA maintains linear complexity in theory, the method introduces an additional term due to multi-head block mixing. This overhead may be negligible in small-scale settings like image classification or ARC-c, but it could become significant in truly long-sequence scenarios. Without explicit runtime or memory benchmarks, the paper fails to demonstrate that MHLA preserves an effective balance between accuracy and computational cost under long-context conditions.
> >
>
> We thank the reviewer for the comment. However, we would like to highlight that we have already conducted the throughput analysis at **Fig. 1b and Fig. 1d** in the main paper.
>
> 1. **Throughput analysis in the main paper.**
>
> We would like to clarify that the paper **indeed** provides throughput benchmarks, specifically in **Fig. 1b** and **Fig. 1d**. As shown in the benchmark results, MHLA achieves a throughput nearly identical to linear attention across different settings.
>
> 1. **Additional Long-Context Experiments.**
>
> In our response to **W1**, we additionally report MHLA’s empirical performance on both the video generation benchmark (Vbench) and the NLP LongBench. For VBench evaluation, MHLA is tested under a sequence length of 31,500 (head number M set to 105, satisfying $M^2 < N$). As shown in the table below, MHLA achieves a **2.1× speedup** over FlashAttention (the same speedup as vanilla linear attention) while also attaining a comparable score on VBench.
>
> On the NLP LongBench, MHLA also delivers results **comparable to current SOTA methods**. These findings collectively demonstrate MHLA’s strong expressivity and efficiency in long-context scenarios.
>
> | Method | Quality score$\uparrow$ | Semantic score$\uparrow$ | Total$\uparrow$ | Latency$\downarrow$ |
> | --- | --- | --- | --- | --- |
> | Wan2.1 1.3B | 85.23 | 75.65 | 83.31 | 166s |
> | Full MHLA | 84.26 | 76.16 | 82.62 | 81s |
> | Full Linear  | 69.96 | 11.38 | 58.24 | 82s |
> | MHLA Hybrid 2/3 | 84.87 | 79.59 | 83.82 | 103s |
>
> *Note: “Full MHLA” and “Full Linear” indicate changing all flash attention to MHLA / vanilla linear attention. “Hybrid 2/3 MHLA” means changing 2/3 layers to MHLA. Both MHLA and vanilla linear attention are accelerated with torch.compile.*

---

> ### Author Response · Authors · 2025-11-23
> **Response to Reviewer 2bkY (4/4)**
>
> **W4: Weak Literature Review and Terminology Issues.**
>
> > The paper’s literature review is notably shallow. The introduction and related work sections lack a comprehensive discussion of the extensive prior research on linear attention, a highly active field with numerous variants and analyses. The authors should consult and reference existing survey papers on linear attention to position MHLA more accurately within the broader landscape. Additionally, the writing style introduces several non-standard or self-coined terms—such as “adapt to each query individually” and “KV summary”—which are not widely adopted in the community.
> >
>
> 1. **Existed Complete Discussion on Prior Research Works**
>
> We appreciate the reviewer’s concern. However, we would like to clarify that a comprehensive analysis of existing linear-attention methods is already provided in **Appendix A**, where we review representative variants and their design motivations in detail. If deemed beneficial, we are willing to incorporate part of the appendix analysis into the main text in the revised version.
>
> 1. **Clarification on Terminology and Conceptual Distinctions**
>
> We would like to clarify that the usage of our terms is indeed different from previous works’ terms from the aspect of implementation and insight. Here are the detailed explanations:
>
> These terms describe **new computational behaviors that do not exist in prior linear-attention formulations**, and therefore cannot be accurately conveyed using previously existing terminology. Below, we detail the two concepts in question, which will be added in the revision for better clarification.
>
> **a. “Adapt to each query respectively”**
>
> This phrase is deliberately used to describe a mechanism **absent from previous linear-attention models**.
>
> Specifically, it refers to the process where:
>
> - **Each query token** is associated with **its own vector of mixing coefficients**
> - These coefficients **weight and aggregate all local KV summaries** independently for every query position.
>
> Thus, the adaptation occurs **per query**, not globally or recurrently, **which is fundamentally different from standard linear attention,** where a fixed recurrence determines the representation for all queries.
>
> **b. “KV summary” (conceptually distinct from the “hidden states” in linear attention)**
>
> The term “KV summary” is indeed different from the concept of “hidden states”. Although the term “KV summary’’ seemingly resembles the “hidden states”, the underlying computation is **structurally different:**
>
> |  | Traditional Linear Attention  | Multi-Head Linear Attention (MHLA) |
> | --- | --- | --- |
> | Dependency among hidden states | **Strict Dependency Chain:** The hidden state at step t $h_t$  strictly depends only on the previous hidden state $h_{t-1}$ via recursive accumulation. No $h_t$  is independent. | **Independent Calculation:** No Global KV Summary $S_g$ depends on any other Global KV Summary. There is no propagation of state across positions. |
> | Hidden states aggregation mechanism | **One-to-One Aggregation:** $h_t$  is computed from a single source (the prior state $h_{t-1}$) using a simple aggregation. | **Many-to-One Aggregation:** Each $S_g$ is computed independently from all local KV summaries using block-specific mixing coefficients. |
>
> As a result, MHLA provides greater expressivity by allowing more diverse and flexible KV summaries compared to the fixed hidden states used in conventional linear attention. We therefore introduce the term **“KV summary’’** specifically to highlight this independence and to avoid conflating MHLA’s behavior with the recurrence-based hidden states in prior work.
>
> In summary, these terms are essential for accurately describing **new mechanisms introduced by MHLA** that have no direct analogue in existing linear-attention formulations. We hope this clarification helps reviewers clearly recognize the conceptual distinctions. We would add a more detailed explanation for the new concepts in the revision.

---

> > ### Comment · Reviewer_2bkY · 2025-11-24
> >
> > The rebuttal satisfactorily addresses my concern and demonstrates that MHLA has potential on longer sequences.
> > I am willing to increase my score to 6.
> > Best of luck.

---

> ### Author Response · Authors · 2025-11-24
>
> Thank you very much for your kind words and for revising the score. We appreciate your time and guidance throughout the review process.

---

### Official Review · Reviewer_uTES · 2025-10-30

**Soundness:** 3
**Presentation:** 3
**Contribution:** 3
**Rating:** 6
**Confidence:** 3

**Summary:**

The paper identifies a failure mode of many kernelized/linear-attention variants namely “global context collapse,” where compressing all keys/values into a single global summary limits rank and drives attention toward near-uniform distributions as sequence length grow .

Autores proposes Multi-Head Linear Attention (MHLA), which partitions tokens into non-overlapping blocks and learns a block-wise mixing of local KV summaries via a nonnegative, normalized coefficient matrix, restoring query-conditioned diversity while keeping linear-time scaling . The authors derive rank bounds showing MHLA can achieve substantially higher attainable rank than global linear attention and empirically observe lower entropy (sharper sparsity) of attention maps .

**Strengths:**

- Clear diagnosis of “global context collapse” with complementary theoretical indicators, e.g. rank upper bound ≤d for global linear attention; MHLA’s additive blockwise rank potential;  and empirical entropy analyses .


- Simple, hardware-friendly construction: blockwise summaries + learned nonnegative mixing; retains linear-time leading term and is compatible with chunkwise/streaming execution .


- Strong cross-domain results: (i) ImageNet improvements over linear attention baselines and competitive with transformer/mamba-style models, (ii) sizable FID gains across DiT scales (including XL) with near-linear-attention throughput, and (iii) modest yet positive NLP benchmarks without positional embeddings .


- Useful ablations: initialization vs learnable mixing and head-number sensitivity give practical guidance (e.g., M=16 works well) .

**Weaknesses:**

- Results are mostly single-number comparisons; there is no reporting of multiple seeds, confidence intervals, or significance tests. For diffusion FID/IS/sFID and ImageNet top-1, please consider report mean±std over ≥3 seeds and specify sample counts and evaluation protocols used for FID/IS (e.g., 50k samples, classifier, resize method) to support the claims of consistent improvements .


- Language modeling evaluation is narrow. The 0.3–0.34B model trained on 5B tokens is assessed on a small suite (ARC-c, Wino, CoPA). Please add perplexity on standard corpora and long-context stress tests to directly evaluate the purported benefits at long sequence lengths, and compare to strong linear/SSM baselines under matched budgets .


- Complexity & memory overhead clarity. The method introduces an M×M mixing step with O(M²d²) cost. While argued negligible when M²≤N, the paper would benefit from explicit profiling of wall-clock time and peak memory versus N and M across hardware and tasks, and from reporting the realized M values used in each experiment (classification/generation/NLP) .


- Streaming/causal claims are not empirically demonstrated. The text asserts compatibility with chunkwise parallel training and streaming/stateful execution; please provide experiments demonstrating long-context streaming inference throughput/latency versus linear attention baselines under causal masking .


- Comparisons breadth in T2I/C2I. The DiT/DiG comparisons are strong, but additional comparisons to other recent linear or hybrid attention mechanisms in diffusion backbones would strengthen the case, especially where prior methods also raise rank or sparsify attention (the paper cites some but not all are evaluated directly) .

Additionally, there are minor typos (e.g., “diversty”, “Iamge”).

**Questions:**

How are the mixing coefficients parameterized to ensure nonnegativity and row normalization in practice (softmax per row, or other), and how sensitive are results to this choice and to the locality-biased initialization? What values of M were used per model and task, and how does wall-clock time and peak memory scale with N and M on H100 across DeiT/DiT/NLP? Please include profiles showing the point where O(M²d²) becomes non-negligible .


Also, can you provide multi-seed runs (≥3) with mean±std for ImageNet accuracy and DiT FID/IS/sFID, including sample counts and evaluation details?

---

> ### Author Response · Authors · 2025-11-23
> **Response to Reviewer uTES (1/4)**
>
> **W1 & Q3: Results are mostly single-number comparisons.**
>
> > There is no reporting of multiple seeds, confidence intervals, or significance tests. For diffusion FID/IS/sFID and ImageNet top-1, please consider report mean±std over ≥3 seeds and specify sample counts and evaluation protocols used for FID/IS (e.g., 50k samples, classifier, resize method) to support the claims of consistent improvements.
> >
>
> We thank the reviewer for the thoughtful comment. Reporting only a single FID/sFID/IS/ImageNet top-1 value follows the common practice established in image classification and image-generation work (e.g., DiG [1], SANA [2], MALA [3]). In the paper, we adopted the same convention to ensure a fair and consistent comparison. However, we also agree with the reviewer’s concern. So, we have provided more detailed experimental settings and additionally report multi-seed results for Image Generation tasks using three different random seeds (0, 42, 100). For image-classification tasks, we tested multiple seeds and observed **no measurable variation** in top-1 accuracy (within 0.01%), which is consistent with the results reported in our paper.
>
> **Experiment setting**. For FID/IS evaluation in the DiT experiments, we use 50k samples, with the Inception-V3 as the classifier, and the method of image pre-processing was set to be bicubic, following the common practice of ImageNet-1k image-generation evaluation.
>
> **Multi-seed results**.
>
> | Model | Resolution | IS (mean ± std) $\uparrow$ | FID (mean ± std) $\downarrow$ | sFID (mean ± std) $\downarrow$ |
> | --- | --- | --- | --- | --- |
> | DiT-S/2 with MHLA | 256x256 | 23.657 ± 0.156 | 59.744 ± 0.100 | 9.811 ± 0.249 |
> | DiT-S/2 with MHLA | 512x512 | 18.965 ± 0.148 | 78.275 ± 0.312 | 13.540 ± 0.305 |
> | DiT-B/2 with MHLA | 256x256 | 39.002 ± 0.170 | 37.519 ± 0.039 | 7.143 ± 0.149 |
> | DiT-L/2 with MHLA | 256x256 | 63.429 ± 0.422 | 21.426 ± 0.051 | 5.690 ± 0.084 |
> | DiT-XL/2 with MHLA | 256x256 | 69.315 ± 0.562 | 19.164 ± 0.031 | 5.628 ± 0.057 |
> | DiT-XL/2 with MHLA + CFG | 256x256 | 251.843 ± 0.359 | 2.586 ± 0.038 | 4.705 ± 0.036 |
>
> [1] Dig: Scalable and efficient diffusion models with gated linear attention. CVPR 2025.
>
> [2] Sana: Efficient high-resolution image synthesis with linear diffusion transformers. *ICLR* 2025.
>
> [3] Rectifying magnitude neglect in linear attention. *ICCV* 2025.

---

> ### Author Response · Authors · 2025-11-23
> **Response to Reviewer uTES (2/4)**
>
> **W2: Language modeling evaluation is narrow.**
>
> > The 0.3–0.34B model trained on 5B tokens is assessed on a small suite (ARC-c, Wino, CoPA). Please add perplexity on standard corpora and long-context stress tests to directly evaluate the purported benefits at long sequence lengths, and compare to strong linear/SSM baselines under matched budgets.
> >
>
> Thank you for your helpful comment. We increase the training token budget and train the 340M model with 10B tokens. We also evaluate it on a wider set of benchmarks (including common-sense reasoning tasks, MMLU, and LongBench) to thoroughly verify the effectiveness of MHLA under various sequence lengths.
>
> Besides, we also compared our method with more recent SOTA methods such as Gated DeltaNet (GDN) [1] and Mamba2 [2]. The result is presented in the tables below. MHLA demonstrates comparable performance with SOTA methods in common-sense reasoning tasks and scores highest in MMLU. It is worth noting that MHLA even surpasses current SOTAs in terms of average score on LongBench.
>
> Lastly, we sincerely hope that the reviewer could take into account the performance advantages of our method in computer vision and generative tasks.
>
> | Tasks | Computer Vision | Vision Generation | NLP |
> | --- | --- | --- | --- |
> | GLA (340M) | NO | NO | Yes |
> | Transformer++(340M) | Yes | Yes | Yes |
> | Mamba* (390M) | NO | NO | Yes |
> | Mamba2* (340M) | NO | NO | Yes |
> | GDN (360M) | NO | NO | Yes |
> | MHLA (340M) | Yes | Yes | Yes |
>
> *Note: Cross-Domain Applicability of MHLA and Other Baselines. **Note:** While Mamba-based architectures can be adapted for vision tasks through auxiliary modules (e.g., VisionMamba), they lack **native support** for non-causal or 2D structures compared to the Transformer family and MHLA.*
>
> | Tasks | MMLU acc $\uparrow$ | **CSR .avg score** $\uparrow$ | Wino acc $\uparrow$ | PIQA acc $\uparrow$ | ARC-c acc_n $\uparrow$ | OBQA acc_n $\uparrow$ | ARC-e acc_n $\uparrow$ | BoolQA acc $\uparrow$ | Wiki ppl $\downarrow$ | LMB ppl $\downarrow$ |
> | --- | --- | --- | --- | --- | --- | --- | --- | --- | --- | --- |
> | GLA (340M) | 22.9 | 46.0 | 50.0 | 62.9 | 25.5 | 31.0 | 45.8 | 60.8 | 41.47 | 86.98 |
> | Transformer++(340M) | 22.9 | 46.8 | 49.6 | 64.4 | 25.7 | 32.8 | 48.1 | 60.5 | 34.57 | 60.46 |
> | Mamba (390M) | 23.5 | 46.4 | 50.5 | 64.1 | 24.9 | 32.4 | **48.3** | 58.2 | 38.32 | 62.43 |
> | Mamba2 (340M) | 23.0 | 47.0 | 49.8 | **64.6** | 25.5 | 32.0 | 49.2 | 61.2 | 35.40 | 58.51 |
> | GDN (360M) | 23.0 | 46.9 | **51.3** | 64.5 | 25.4 | 31.4 | 47.3 | **62.0** | 35.01 | 60.16 |
> | MHLA (340M) | **23.7** | **47.1** | **51.3** | 64.4 | **25.9** | **33.4** | 46.5 | 61.3 | 38.31 | 71.64 |
>
> *Note: Comparison of common-sense reasoning tasks and MMLU.*
>
> |  | Overall Performance | Multi-Doc QA |  |  | summarization |  |  | code |  | Single-Doc QA |  | Few-shot |  | Synthetic Task |  |
> | --- | --- | --- | --- | --- | --- | --- | --- | --- | --- | --- | --- | --- | --- | --- | --- |
> | Model | avg $\uparrow$ | hotpotqa $\uparrow$ | musique $\uparrow$ | 2wikimqa $\uparrow$ | qmsum $\uparrow$ | gov_report $\uparrow$ | multi_news $\uparrow$ | repobench-p $\uparrow$ | lcc $\uparrow$ | qasper $\uparrow$ | nqa $\uparrow$ | samsum $\uparrow$ | triviqa $\uparrow$ | passage retrieval en $\uparrow$ | passage retrieval zh $\uparrow$ |
> | Mamba | *6.97* | 2.36 | 1.60 | 3.37 | 12.23 | *18.36* | 14.96 | **13.63** | 12.33 | 4.57 | 2.28 | 5.16 | 5.49 | 1.1 | 0.1 |
> | GLA | 6.53 | 2.31 | 1.67 | 3.23 | 11.42 | 17.72 | 15.34 | *13.59* | 12.55 | 4.53 | 2.13 | 3.94 | 0.7 | **1.98** | 0.27 |
> | GDN | 6.86 | 2.24 | 1.54 | 2.86 | 12.46 | 17.91 | **15.98** | 10.42 | 9.98 | **4.73** | **2.48** | **6.85** | **7.61** | 0.53 | 0.41 |
> | Transformer++ | 6.92 | 2.13 | **2.22** | **4.97** | 11.75 | 16.81 | *15.11* | 11.56 | 9.92 | 4.45 | 2.35 | 6.24 | *7.47* | 0.76 | 1.18 |
> | Mamba2 | 6.62 | *2.38* | 1.69 | 3.56 | *12.57* | 17.65 | 14.00 | 10.15 | 9.49 | *4.70* | 2.20 | 4.97 | 7.03 | 0.72 | **1.51** |
> | MHLA | **7.41** | **2.97** | *1.87* | *3.58* | **12.58** | **18.59** | 15.01 | 13.37 | **12.72** | 4.68 | *2.38* | *6.41* | 6.44 | *1.69* | *1.49* |
>
> *Note: Comparison on LongBench.*
>
> [1] Gated Delta Networks: Improving Mamba2 with Delta Rule. ICLR 2025.
>
> [2] Transformers are SSMs: Generalized Models and Efficient Algorithms Through Structured State Space Duality. ICML 2024.

---

> > ### Author Response · Authors · 2025-11-23
> > **Response to Reviewer uTES (3/4)**
> >
> > **W3 & Q2: Complexity & memory overhead clarity.**
> >
> > > The method introduces an M×M mixing step with O(M²d²) cost. While argued negligible when M²≤N, the paper would benefit from explicit profiling of wall-clock time and peak memory versus N and M across hardware and tasks, and from reporting the realized M values used in each experiment.
> > >
> >
> > We thank the reviewer for the thoughtful suggestion. The value of M for each main experiment in the paper was reported in **Section 4.1, line 356, and Section 4.2, line 388**.
> >
> > The Language Modeling experiments’ setting has been added in the revision, where we set $M$ to 32, which also satisfies $M^2 < N$ (the context length is set to 2048). Following the reviewer’s suggestion, we perform explicit profiling of the latency and memory costs under different sets of $N$ and $M$. The result is presented in the table below, showing that when $M^2$ is lower or about equal to the value of $N$, the extra cost brought by “Multi-Head Mixing” is **negligible.**
> >
> > | M/N | 256 | 1024 | 4096 |
> > | --- | --- | --- | --- |
> > | 4 | 42ms 3.7G | 52ms 7.1G | 147ms 20.8G |
> > | 16 | 40ms 3.9G | 51ms 7.2G | 145ms 21G |
> > | 64 | 39ms 4.8G | 52ms 8G | 148ms 21.7G |
> > | 256 | — | 61ms 12G | 157ms 25.4G |
> > | 1024 | — | — | 219ms 40G |
> >
> > *Note: Profiling result on DiT-S/2 on an A100 GPU.*
> >
> > | M/N | 256 | 1024 |
> > | --- | --- | --- |
> > | 4 | 129 imgs/s 3.4G | 124 imgs/s 8.9G |
> > | 16 | 118 imgs/s 3.8G | 118 imgs/s 9.4G |
> > | 64 | 150 imgs/s 5.7G | 104 imgs/s 11G |
> > | 256 | — | 89 imgs/s 18G |
> >
> > *Note: Profiling result on DeiT-S/16 on a V100 GPU.*
> >
> > | M/N | 2048 | 4096 | 8192 |
> > | --- | --- | --- | --- |
> > | 16 | 11100 tokens/s 4.83G | 22000 tokens/s 7.46G | 40500 tokens/s 13.04G |
> > | 32 | 11000 tokens/s 4.99G | 21800 tokens/s 7.55G | 39700 tokens/s 12.84G |
> > | 64 | — | 21100 tokens/s 7.89G | 39400 tokens/s 13.06G |
> > | 128 | — | — | 29600 tokens/s 13.79G |
> >
> > *Note: Profiling results for the language model are obtained on an H100 GPU. We report the theoretically derived values, aligned with empirical measurements, and we believe that further engineering optimizations will follow the same trend and achieve stronger performance.*
> >
> > **w4: Streaming/causal claims are not empirically demonstrated.**
> >
> > > The text asserts compatibility with chunkwise parallel training and streaming/stateful execution; please provide experiments demonstrating long-context streaming inference throughput/latency versus linear attention baselines under causal masking.
> > >
> >
> > We thank the reviewer for the helpful comment. To validate MHLA’s compatibility with chunkwise-parallel training, we measure the training throughput under various sequence lengths, and the results are in the table below.
> >
> > It is clearly observed that MHLA has the highest throughput under the “**w/o optimized kernel**” setting. Further optimization with kernel fusion tricks could significantly improve the throughput. However, due to the time limitation, we have not implemented it yet. For your reference, we have put the throughput for GLA and GDN with kernel optimization tricks to show the speed-up effect, where both of methods have achieved around 3x speedup. This can be used to infer the performance of MHLA, which could also achieve 3x speedup.
> >
> > | Type | Model | 2K x 16 | 4K x 8 | 8K x 4 | 16K x 2 |
> > | --- | --- | --- | --- | --- | --- |
> > | pytorch implementation **w/o optimized kernel**  | MHLA | 83249 | 83677 | 81762 | 81427 |
> > |  | GDN | 28763 | 19217 | 12306 | 7262 |
> > |  | GLA | 57815 | 31875 | 18092 | 17375 |
> > | pytorch implementation **w/ optimized kernel**  | GDN | 81437 (2.8x) | 81596 | 80739 | 78049 |
> > |  | GLA | 141386 (2.5x) | 139735 | 139871 | 137635 |
> > |  | Mamba2 | 96871 | 96787 | 96911 | 96450 |
> >
> > *Note: Training throughput on H100. We report tokens/s under a different set of “sequence length x batch size”.*
> >
> > To further address the reviewer’s concerns regarding inference latency, we additionally benchmark several baselines, reported in the table below. Notably, our PyTorch implementation of MHLA is the fastest among “w/o optimized kernel” setting, and **even surpasses the heavily optimized Mamba2**.
> >
> > | Type | Model | Latency |
> > | --- | --- | --- |
> > | pytorch implementation **w/o optimized kernel**  | MHLA | 204.3s |
> > |  | GLA | 269.4s |
> > |  | GDN | 292.3s |
> > | pytorch implementation **w/ optimized kernel**  | GDN | 126.8s |
> > |  | GLA | 144.4s |
> > |  | Mamba2 | 311.0s |
> >
> > *Note: Latency tested on H100 with sequence length to 4096 tokens. MHLA‘s head number M is set to 32.*

---

> > > ### Author Response · Authors · 2025-11-23
> > > **Response to Reviewer uTES (4/4)**
> > >
> > > **w5: Comparisons breadth in image generation evaluations.**
> > >
> > > > The DiT/DiG comparisons are strong, but additional comparisons to other recent linear or hybrid attention mechanisms in diffusion backbones would strengthen the case, especially where prior methods also raise rank or sparsify attention (the paper cites some but not all are evaluated directly).
> > > >
> > >
> > > We thank the reviewer for the helpful suggestion. We also agree that adding more recent works will further strengthen the paper. However, we would like to highlight that our main experiments and ablation experiments are on image classification, where we have run most of the experiments. And then, after confirming the training recipe and model architecture, we verify it on vision generative and diffusion tasks, which is why it is not as comprehensive as the vision ablation. However, following the reviewer’s suggestion, we will add more ablations under diffusion tasks in the revision.
> > >
> > > Thus, we have included LiT [3], a work that conducts image-generation experiments on DiT, as a strong baseline. The results show that **MHLA continues to outperform existing methods, further demonstrating its effectiveness.**
> > >
> > > | Model | FID (mean ± std) $\downarrow$ |
> > > | --- | --- |
> > > | LiT-S/2 | 63.21 |
> > > | DiT-S/2 with MHLA | 59.744 ± 0.100 |
> > > | LiT-B/2 | 40.86 |
> > > | DiT-B/2 with MHLA | 37.519 ± 0.039 |
> > > | LiT-L/2 | 24.04 |
> > > | DiT-L/2 with MHLA | 21.426 ± 0.051 |
> > > | LiT-XL/2 | 20.66 |
> > > | DiT-XL/2 with MHLA | 19.164 ± 0.031 |
> > >
> > > [1] Breaking the Low-Rank Dilemma of Linear Attention. CVPR 2025.
> > >
> > > [2] Rectifying magnitude neglect in linear attention. ICCV 2025.
> > >
> > > [3] LiT: Delving into a Simple Linear Diffusion Transformer for Image Generation. ICCV 2025.
> > >
> > > **w6: Typos**
> > >
> > > > Additionally, there are minor typos (e.g., “diversty”, “Iamge”).
> > > >
> > >
> > > Thank you for pointing out the minor typos (e.g., “diversty”, “Iamge”). We will correct all of them in the revised version.
> > >
> > > **Q1: About initialization and normalization of the mixing coefficients.**
> > >
> > > > How are the mixing coefficients parameterized to ensure nonnegativity and row normalization in practice (softmax per row, or other), and how sensitive are results to this choice and to the locality-biased initialization?
> > > >
> > >
> > > Thanks for the comment. In practice, we clip the mixing coefficients to the range (0, 1) at every training step to ensure non-negativity and maintain stability. We do not apply explicit normalization for efficiency. Empirically, this simple clipping operation keeps the **row-wise sums of the mixing coefficients inherently stable**, effectively yielding an approximate row-normalization behavior. Without this clipping, training quickly encounters a gradient explosion and fails in the early stage.
> > >
> > > The ablation study on locality-biased initialization is provided in **Tab. 6a** of the original paper, illustrating the sensitivity of the results to different initialization schemes. We copied the table below.
> > >
> > > | Local-biased Initialization | Learnable | Top1-acc (%) |
> > > | --- | --- | --- |
> > > | no | yes | 75.4 |
> > > | yes | no | 75.1 |
> > > | yes | yes | 75.8 |
> > >
> > > *Note: Ablation result on DeiT-T-MHLA.*

---

> ### Comment · Reviewer_uTES · 2025-11-23
> **Why FlashAttention is worse than MHLA on Memory Overhead?**
>
> I thank the authors for their rebuttal, which helped clarify several aspects of the paper. After re-reading the manuscript, I have an additional question regarding Fig. 1(a).
>
> In the figure, FlashAttention is reported as out-of-memory (OOM) at high resolutions, while MHLA continues to run. Since FlashAttention is explicitly designed to avoid materializing the (N^2) attention matrix and has (O(Nd)) activation memory, I would expect it to exhibit memory behavior similar to linear attention methods, as in both cases the largest tensors materialized in HBM are typically the Q/K/V matrices of size (O(Nd)).
>
> Could the authors clarify why FlashAttention encounters OOM under this setting while MHLA does not?
>
> Additionally, could the authors report the peak allocated GPU memory for both FlashAttention and MHLA in these experiments, to help determine whether the observed OOM behavior arises from fundamental complexity differences or from implementation- or caching-related factors?

---

> ### Author Response · Authors · 2025-11-24
> **Response to Reviewer uTES**
>
> Q5: Memory comparison between Flash Attention and MHLA
>
> >  After re-reading the manuscript, I have an additional question regarding Fig. 1(a).
> >
> >
> > In the figure, FlashAttention is reported as out-of-memory (OOM) at high resolutions, while MHLA continues to run. Since FlashAttention is explicitly designed to avoid materializing the (N^2) attention matrix and has (O(Nd)) activation memory, I would expect it to exhibit memory behavior similar to linear attention methods, as in both cases the largest tensors materialized in HBM are typically the Q/K/V matrices of size (O(Nd)).
> >
> > Could the authors clarify why FlashAttention encounters OOM under this setting while MHLA does not?
> >
> > Additionally, could the authors report the peak allocated GPU memory for both FlashAttention and MHLA in these experiments, to help determine whether the observed OOM behavior arises from fundamental complexity differences or from implementation- or caching-related factors?
> >
>
> Thank you very much for your thoughtful comment. We would like to kindly clarify two points for better understanding: (1) Fig. 1(a) shows the generation results of our fine-tuned model, and we believe the comment might have been referring to Fig. 1(d); and (2) the OOM indicator is meant for self-attention instead of flash-attention, **as it is not supported on V100 GPU**.
>
> In Fig. 1(d), we use the color coding for “OOM”  to indicate its correspondence to self-attention (both with **red-pink** color), whereas FlashAttention results on other devices are shown in **light blue**. To make it clearer, we have refined the figure with an NA label for flash attention on V100.
>
> Besides, we also agree with the reviewer that flash-attention should have similar memory consumption behaviour as linear attention, and the results are shown in the table below. It is observed that the flash-attention even consumes less memory, compared to linear attention. We thought this was due to the highly optimized kernel implementation of flash-attention.
>
> |                | Flash Attention | Linear Attention | MHLA |
> |----------------|-----------------|------------------|------|
> | **Peak GPU Memory** | 18G            | 19G             | 22G  |

---

### Official Review · Reviewer_WAkW · 2025-11-01

**Soundness:** 3
**Presentation:** 2
**Contribution:** 2
**Rating:** 2
**Confidence:** 4

**Summary:**

This paper raises a failure mode in linear attention, which the authors term "global context collapse." They argue that compressing the entire sequence into a single, shared key-value summary leads to a severe bottleneck in representational capacity, which they quantify through rank and entropy analysis of the attention matrix. To address this issue, this paper proposes Multi-Head Linear Attention (MHLA), a new mechanism that partitions the input sequence along the token dimension into multiple blocks. MHLA computes local summaries for each block. For each query block, it then constructs a query-conditioned context by computing a learnable, weighted mixture of all local summaries. The authors validate MHLA across diverse domains, including image classification, image generation, and natural language processing, demonstrating competitive performance gains over baselines.

**Strengths:**

1. The diagnosis of "global context collapse," supported by a concise analysis of rank deficiency and entropy elevation, provides intuitive motivation for the work. It clearly articulates why previous linear attention models often underperform.
2. The paper presents extensive experiments across multiple domains (computer vision and NLP).

**Weaknesses:**

1. The choice of the term "Multi-Head" is confusing and conflicts with the well-established definition from original Transformer, which refers to splitting the channel dimension. In this paper, "heads" are defined along the token/spatial dimension. This non-standard usage could lead to significant confusion in the community.
2. The paper misses some important comparisons with highly relevant prior work such as FLASH[1] and VOLO[2], which also employs a block-wise strategy combining quadratic and linear attention. For example, VOLO-D1 (comparable to DeiT-S) achieves 84.2 on ImageNet, which substantially higher than the MHLA's 81.0. It is worth noting that VOLO-D1 was proposed four years ago.


[1] Transformer Quality in Linear Time, ICML 2022

[2] VOLO: Vision Outlooker for Visual Recognition TPAMI

**Questions:**

1. How does MHLA compare, conceptually and empirically, to other block-wise attention strategies? For instance, how does its "soft mixing" of all block summaries contrast with the approach in FLASH, which uses a hybrid of quadratic intra-block and linear inter-block attention?
2. The authors report the “Throughput of DiT-S/2 with 4096 resolution” in Figure 1(d), but no experimental generation results at 4096×4096 resolution are provided in the paper. Presenting throughput for a resolution that was not actually evaluated risks implying overclaimed contribution and should be supported with real experiments.

---

> ### Author Response · Authors · 2025-11-20
> **Response to Reviewer WAkW (1/2)**
>
> We sincerely appreciate your comments.
>
> **W1: Confusion regarding the term "Multi-Head"**
>
> > The choice of the term "Multi-Head" is confusing and conflicts with the well-established definition from original Transformer.
> >
>
> Thank you for pointing out the potential confusion with the naming convention. However, we would like to highlight that we have used a clear attribute “Token-level” to differentiate our method from the conventional MHSA. Besides, our proposed neural operator lies in the category of linear attention, which has been well known to be different from the self-attention operators. Also, we believe the name “Token-level MHLA” best describes the concept of our proposed method, as it indeed divides the tokens into different `heads'.  If the reviewer still believes there is a risk of confusion, we could change the name in the revision.
>
> **W2 & Q1: Missing comparisons with VOLO and FLASH**
>
> > The paper misses some important comparisons with highly relevant prior work such as FLASH[1] and VOLO[2]. For example, VOLO-D1 (comparable to DeiT-S) achieves 84.2 on ImageNet, which substantially higher than the MHLA's 81.0.
>
> > How does MHLA compare, conceptually and empirically, to other block-wise attention strategies?
>
>
> Thanks for your comment. We believe the reviewer is using a wrong result compared with VOLO, and we argue that Token-level MHLA is significantly different from FLASH  and VOLO in terms of both concepts, method, and performance.
>
> **1. Clear Accuracy Advantage over VOLO**
>
> We thank the reviewer for the comment. **However, we believe the reviewer is using the wrong result for the comparison with the newly proposed VOLO baseline by the reviewer.** The results that the reviewer referenced are from the ablation parts with simplified training settings to speed up the experiment on the impact of Token-level MHLA. It is unfair to use our “toy” results to compare with the full training results in VOLO. Actually, it is clearly shown in Tab. 2b that our method achieves 0.4% higher accuracy with 30% less FLOPs over the VOLO-D1 model.
>
> More importantly, image classification is only one of the three tasks. To prove the generalization capability of the proposed method, we have also run it on image generation and NLP tasks, where, among all the tasks, our method has clear advantages over the selected baseline.
>
> **2. Contribution Difference Between VOLO and Token-level MHLA**
>
> We argue that VOLO’s *outlooker* module is **significantly architecturally different** from both block-wise attention and MHLA’s multi-head mixing. We would like to highlight that the two methods lie in different categories. VOLO is more similar to a convolution operation with dynamic weights. Differently, Token-level MHLA lies in the category of linear attention with improved expressivity through a new token-level head division.
>
> **3. Conceptual Difference Between FLASH and our MHLA**
>
> MHLA  fundamentally differs from FLASH's mixed chunk attention:
>
> First, FLASH does not have inter-block communication. Differently, Token-level MHLA implements a dedicated design to boost both the inter-block and thus achieves significantly better results, as shown in the next parts (Sec. 3.2 and Sec. 3.3) and the experimental comparison in the fourth part of this response.
>
> Secondly, Token-level MHLA has been verified across diverse tasks, including computer vision, generative tasks, and NLP tasks. In contrast, FLASH is only verified on NLP tasks. A table summarization is shown below for your fast reference.
>
> |  | MHLA | FLASH |
> | --- | --- | --- |
> | Architecture & Implementation | Multi-Head Mixing realizes fine-grained inter-block interaction. | No inter-block interaction. |
> | Tasks & Modalities | Generalizability across CV, Image Generation, Language Modeling tasks and so on. | Only validated on Language Modeling. |
> **4. Method and Performance Advantages over FLASH**
>
> As mentioned above, FLASH was primarily designed for NLP tasks. To further verify the advantages of Token-level MHLA over FLASH, we conducted additional experiments in language modeling and report the results in the table below. It is clearly observed that Token-level MHLA achieves significant improvements over FLASH over all tasks.
>
> | Tasks | Wiki ppl $\downarrow$ | LMB ppl $\downarrow$ | PIQA acc $\uparrow$ | Wino acc $\uparrow$ | ARC-e acc_n $\uparrow$ | ARC-c acc_n $\uparrow$ | OBQA acc_n $\uparrow$ | BoolQA acc $\uparrow$ | CSR .*avg* $\uparrow$ | MMLU acc $\uparrow$ |
> | --- | --- | --- | --- | --- | --- | --- | --- | --- | --- | --- |
> | MHLA (325M) | **38.31** | **71.64** | **64.4**  | **51.3** | **46.5** | **25.9** | **33.4** | **61.3** | **47.1** | **23.7** |
> | FLASH (325M) | 65.3 | 123.6 | 59.4 | 50.9 | 43.4 | 24.5 | 29.8 | 61.1 | 44.9 | 22.1 |
>
> *Note: In the table above, we control the total parameters of all models to around 340M and train the models with 10B tokens from FineWeb-Edu.*

---

> ### Author Response · Authors · 2025-11-20
> **Response to Reviewer WAkW (2/2)**
>
> **5. Summary**
>
> We appreciate your suggestions to include these baselines. However, we believe the reviewer referred to the wrong result for the comparison with VOLO on the classification task. Experimental results in Tab. 2b clearly show that Token-level MHLA achieves better accuracy with less computation than VOLO.
>
> Besides, Token-level MHLA surpasses FLASH and VOLO in their own fields, respectively (Image Classification and Language Modeling),  demonstrating the strong architectural **generalizability** of the proposed Token-level MHLA.
>
> More importantly, the method, novelty, and the implementation of token-level MHLA are different from both FLASH and VOLO, demonstrating our contribution. We will add those baselines in the revision.
>
> **Given those facts, we sincerely ask the reviewer to re-rate the paper.** If the reviewer still has any other questions, we are more than happy to have more communication.
>
> **Q2: Throughput for 4096 resolution without generation results**
>
> > The authors report the “Throughput of DiT-S/2 with 4096 resolution” in Figure 1(d), but no experimental generation results at 4096×4096 resolution are provided in the paper. Presenting throughput for a resolution that was not actually evaluated risks implying overclaimed contribution and should be supported with real experiments.
> >
>
> Thank you for your comments. We would like to highlight that we are following previous peer-reviewed papers’ practice in the field of image generation and computer vision (similar practice can be found in DiG [1] (Fig. 2), SANA [2] (Fig. 1), and DiffuSSM [3] (Fig. 3)). If the reviewer still thinks this is confusing, we will highlight this setting in the revision.
>
> [1] Dig: Scalable and efficient diffusion models with gated linear attention. CVPR 2025.
>
> [2] Sana: Efficient high-resolution image synthesis with linear diffusion transformers. ICLR 2025.
>
> [3] Diffusion models without attention. CVPR 2024.

---

> ### Author Response · Authors · 2025-11-24
>
> Dear reviewer WAkW,
>
> We greatly appreciate your time and effort in reviewing our work. We are eager to ensure that we have adequately addressed your concerns and are prepared to offer further clarifications or address any additional questions you may have. We would be grateful if you could share your thoughts on our rebuttal.
>
> Best regards, The Authors

---

> ### Comment · Reviewer_WAkW · 2025-11-25
> **Response to Authors**
>
> Thank you for the detailed rebuttal and the additional experiments. I appreciate the clarifications, but several of my concerns remain.
>
> 1. On the comparison with VOLO
>
> You state that I used a “wrong result” because the 81.0 ImageNet accuracy comes from ablation experiments with simplified training settings (“toy” results). However, in the main paper this is not clearly indicated as a toy setting, and the text describes replacing DeiT’s attention and training for 300 epochs, which reads as a standard setting rather than a strongly downscaled toy example. As a reader, it is therefore natural to treat these numbers as meaningful baselines.
>
> Regarding Tab. 2b, I acknowledge that MHLA-VLT-S slightly outperforms VOLO-D1 by 0.4%. However, this comparison is not entirely symmetric, since MHLA-VLT-S is your strongest model built on a VLT backbone while VOLO-D1 is the weakest in the VOLO family (D1–D5). If the main claim is “clear accuracy advantage over VOLO,” I would expect a more systematic comparison against stronger VOLO variants (e.g. D2 or higher) at similar budgets.
>
>
> 2.	Conceptual relation between MHLA and VOLO
>
> The authors argue that VOLO is closer to dynamic convolution and that Token-level MHLA lies in a different category of linear attention. From my perspective as a reader, both methods can still be viewed as ways of localizing or structuring what would otherwise be global/full attention. VOLO’s outlooker groups tokens in a 2D spatial fashion with local receptive fields. MHLA groups tokens along a 1D structure (blocks) and performs attention within these chunks. So while I agree that the implementation details differ (dynamic conv–like outlooker vs token-level head mixing), conceptually they both reduce global attention to block/local attention patterns. My original question (“How does MHLA compare, conceptually and empirically, to other block-wise attention strategies?”) was aimed at exactly this: positioning MHLA in the broader landscape of local/block-wise attention, including VOLO. I still feel this connection is somewhat downplayed in the current wording, and a more explicit acknowledgement of this shared design space would make the paper clearer and fairer to prior work.
>
>
> 3.	Video generation experiments and comparison to efficient attention baselines
>
> The additional video generation experiments are interesting and potentially a strong point of the paper. However, implementation details are missing: e.g., training data, number of GPUs, training steps, and the exact evaluation benchmarks. Without these, it is difficult to properly assess the strength and practicality of the results.
>
> Moreover, surpassing a “native” linear attention baseline in video generation is not difficult. Vanilla linear attention is known to be a relatively weak choice compared to more specialized efficient-attention or sparse-attention designs. If you want to claim video generation as a key contribution of MHLA, it would be much more convincing to compare against dedicated accelerated video generation methods, such as STA [1] or sparse video generation approaches [2,3,4], under comparable training budgets and evaluation protocols.
>
> 4.  Throughput for 4096 resolution without generation results
>
> Thank you for the clarification and for pointing to prior work such as DiG, SANA, and DiffuSSM. I agree that reporting throughput vs. resolution curves (as in your **Figure 1(b)**) is a reasonable and common practice, and I have no objection to that type of analysis.
> However, my concern is specifically about **Figure 1(d)**, where you highlight the Throughput at 4096×4096 and use this as evidence for acceleration at 4K resolution, without any actual 4096×4096 generation results (qualitative or quantitative) in the paper. This creates a mismatch between what is measured and what is demonstrated, and risks overclaiming the practical impact at that resolution. You mention DiG, SANA, and DiffuSSM as precedent, but these works do not present a figure analogous to your Figure 1(d) that isolates a single, very high resolution (e.g., 4096×4096) as a key claim without also providing corresponding generation results at that resolution. It isworth noting that, **SANA reports both qualitative and quantitative results at multiple resolutions (512², 1024², 2048², 4096²)**, so its 4K claims are supported by actual experiments rather than throughput alone.
>
> [1] Fast Video Generation with Sliding Tile Attention. ICML 2025
>
> [2] Sparse VideoGen: Accelerating Video Diffusion Transformers with Spatial-Temporal Sparsity.  ICLR 2025
>
> [3] Sparse VideoGen2: Accelerate Video Generation with Sparse Attention via Semantic-Aware Permutation NeurIPS 2025
>
> [4] VSA: Faster Video Diffusion with Trainable Sparse Attention. NeurIPS 2025

---

> > ### Author Response · Authors · 2025-11-26
> > **Response to Reviewer WAkW (4)**
> >
> > **Q2.1: Is Token-level MHLA a local attention?**
> >
> > > The authors argue that VOLO is closer to dynamic convolution and that Token-level MHLA lies in a different category of linear attention. From my perspective as a reader, both methods can still be viewed as ways of localizing or structuring what would otherwise be global/full attention.
> > >
> >
> > Thanks for the comment. Actually, MHLA  **is** under the category of **global attention**. As demonstrated in Eq. 4, when all mixing coefficients are set to 1, MHLA will perform exactly the same as vanilla linear attention (which is absolutely a global attention). So it is worth noting that **MHLA is a special variant of global attention**, instead of local/block attention.
> >
> > **Q2.2: Does Token-level MHLA groups tokens along a 1D structure?**
> >
> > > VOLO’s outlooker groups tokens in a 2D spatial fashion with local receptive fields. MHLA groups tokens along a 1D structure (blocks) and performs attention within these chunks.
> > >
> >
> > MHLA is clearly demonstrated as a 2D structure in terms of image classification (see line 202-203 and Fig. 4 in the original paper). It is only 1D in Language Modeling.
> >
> > **Q2.3: Does Token-level MHLA reduce to local attention?**
> >
> > > So while I agree that the implementation details differ (dynamic conv–like outlooker vs token-level head mixing), conceptually they both reduce global attention to block/local attention patterns.
> > >
> >
> > As clarified in our response to Q2.1, MHLA is a **special variant of global attention.** Therefore, MHLA does not reduce to local attention.
> >
> > **Q2.4: Does VOLO lie in the category of Attention?**
> >
> > > My original question (“How does MHLA compare, conceptually and empirically, to other block-wise attention strategies?”) was aimed at exactly this: positioning MHLA in the broader landscape of local/block-wise attention, including VOLO.
> > >
> >
> > According to the VOLO paper, the outlooker **is not categorized as an attention module** (see Tab. 6 of VOLO’s paper); In VOLO’s outlooker module, there are **no concepts of Query, Key, and Value, which are all critical components in the common attention mechanism** and therefore do not align with the widely accepted definition of attention in the community. As explained in the previous response, the outlooker in VOLO is **explicitly designed to capture local information**, operating as a dynamic convolution–style local aggregator.
> >
> > | Model | Layer type | #Params | Top-1 Acc. |
> > | --- | --- | --- | --- |
> > | VOLO-D1 | Outlooker | 27M | **84.2** |
> > | VOLO-D1 | Local self-attention | 27M | 83.8 |
> > | VOLO-D1 | Convolution | 27M | 83.8 |
> >
> > *Note. Tab. 6 in VOLO’s paper, which does not place itself as a local attention.*
> >
> > **Q2.5: Was the relationship between Token-level MHLA and local/block-wise attention downplayed?**
> >
> > > I still feel this connection is somewhat downplayed in the current wording, and a more explicit acknowledgement of this shared design space would make the paper clearer and fairer to prior work.
> > >
> >
> > As explained in the above responses, Token-level MHLA is not a local/block-wise attention but actually a **global attention**. Token-level MHLA divides the sequence into blocks is for globally and comprehensively collect them together. **The KV Summary are indeed for global-wise operation**. It is quite different from the so-called “block-wise” mechanism in VOLO or other works. However, we can add more papers leveraging local / block-wise strategies like VOLO in the reference if the reviewer wants.

---

> > ### Author Response · Authors · 2025-11-26
> > **Response to Reviewer WAkW (6)**
> >
> > **Q4: Throughput for 4096 resolution without generation results**
> >
> > > However, my concern is specifically about **Figure 1(d)**, where you highlight the Throughput at 4096×4096 and use this as evidence for acceleration at 4K resolution, without any actual 4096×4096 generation results (qualitative or quantitative) in the paper. This creates a mismatch between what is measured and what is demonstrated, and risks overclaiming the practical impact at that resolution. You mention DiG, SANA, and DiffuSSM as precedent, but these works do not present a figure analogous to your Figure 1(d) that isolates a single, very high resolution (e.g., 4096×4096) as a key claim without also providing corresponding generation results at that resolution. It is worth noting that, **SANA reports both qualitative and quantitative results at multiple resolutions (512², 1024², 2048², 4096²)**, so its 4K claims are supported by actual experiments rather than throughput alone.
> > >
> >
> > Thank you for the comment. We apologize for the misreference of SANA. However, the assertion that DiG “does not present a figure analogous to Fig. 1d” is not correct. **The Fig. 3d of DiG also shows their models’ speed at a single 2048x2048 resolution without providing corresponding generation results.** Additionally, we have already provided Fig. 1b to clearly present the acceleration trend of the proposed MHLA as resolution grows, preventing it from only showing under a single high resolution. The purpose of Fig. 1(d) is to serve as an extension of Fig. 1(b), demonstrating the general efficiency advantage of MHLA across different hardware.

---

> ### Author Response · Authors · 2025-11-26
> **Response to Reviewer WAkW (3)**
>
> **Summary**
>
> We thank the reviewer for the positive feedback. We are pleased that the reviewer **acknowledged our diagnosis of *global context collapse*** in existing linear attention, as well as **the breadth of our experiments** across multiple domains, and **did not identify any critical weaknesses** in our work. We have broken down the reviewer’s new questions into points to better address the concerns.
>
> **Q1.1: Are the results of Tab. 2a under simplified training settings?**
>
> > You state that I used a “wrong result” because the 81.0 ImageNet accuracy comes from ablation experiments with simplified training settings (“toy” results). However, in the main paper this is not clearly indicated as a toy setting, and the text describes replacing DeiT’s attention and training for 300 epochs, which reads as a standard setting rather than a strongly downscaled toy example. As a reader, it is therefore natural to treat these numbers as meaningful baselines.
> >
>
> Thank you for your comment. In the image classification literature, uniformly **replacing the attention module on DeiT could be treated as an ablation experiment** (see Tab. 2 of MALA [1] and Fig.6/Tab.3 of Flatten Transformer [2]). In contrast, for the state-of-the-art models (including VOLO), tens of tricks besides the core contribution could be used to improve the final accuracy, as shown in the table below:
>
> | Tricks | VOLO-D1 | MHLA-VLT-S (Tab. 2b) | DeiT Ablation Setting (Tab. 2a) |
> | --- | --- | --- | --- |
> | Random augmentation | Yes | Yes | Yes |
> | EMA | Yes | Yes | Yes |
> | Longer training epochs (300) | Yes | Yes | Yes |
> | Architectural modifications - model depth | Yes | Yes | No |
> | More convs for patch embedding | Yes | Yes | No |
> | More transformers (12 → 14) | Yes | No | No |
> | Enhanced residual connection | Yes | No | No |
> | Token labeling with MixToken | Yes | No | No |
> | Increase Heads in Transformer | Yes | No | No |
>
> **Note: (1) All of the above tricks in the table are not VOLO’s core contribution, which is Outlooker module. (2)** *Architectural modifications - model depth indicates adding more outlooker (VOLO) blocks or MHLA blocks based on DeiT-Small.*
>
> The training results in Tab. 2a in the main paper exactly lie in the simplified training category, which we claim as “under a simplified training setting (”toy” results setting)” because it uses the *exact same tricks as DeiT, with only the attention module replaced with MHLA*, ensuring a fair ablation.  This setup is **intentionally designed to isolate the effect of the attention module itself, rather than to pursue competitive accuracy numbers**. We will clarify this point in the revised paper.
>
> [1] Rectifying magnitude neglect in linear attention. *ICCV* 2025.
>
> [2] FLatten Transformer: Vision Transformer using Focused Linear Attention. *ICCV 2023.*
>
> **Q1.2: Asking for the classification performance on a larger model with a new baseline**
>
> > Regarding Tab. 2b, I acknowledge that MHLA-VLT-S slightly outperforms VOLO-D1 by 0.4%. However, this comparison is not entirely symmetric, since MHLA-VLT-S is your strongest model built on a VLT backbone while VOLO-D1 is the weakest in the VOLO family (D1–D5). If the main claim is “clear accuracy advantage over VOLO,” I would expect a more systematic comparison against stronger VOLO variants (e.g. D2 or higher) at similar budgets.
> >
>
> Thank you for your suggestion. However, we want to emphasize that comparing MHLA-VLT-S with VOLO-D1 is fair in terms of both parameters (28M and 27M) and FLOPs (4.6G and 6.8G).
>
> However, to justify that the advantages of MHLA over VOLO could be generalized to larger models, we additionally provide experiments on a larger classification model and compare it against VOLO-D2. Despite having **significantly fewer parameters (−15.3%)** and **lower FLOPs (−29.8%)**, our MHLA-VLT-B still achieves a **higher** Top-1 accuracy.
>
> | Model | params | FLOPs | Top1-ACC |
> | --- | --- | --- | --- |
> | MHLA-VLT-B | 50M | 9.9G | 85.6 |
> | VOLO-D2 | 59M | 14.1G | 85.2 |
>
> *Note: MHLA-VLT-B is trained strictly following the same training settings as described in our paper, and both MHLA-VLT-B and VOLO-D2 are trained for **300 epochs on ImageNet-1K**. All evaluations are conducted on the **ImageNet-1K validation set** to ensure a fair comparison.*
>
> Lastly, we want to highlight that MHLA is a universal attention operator across multiple tasks in CV, vision generation, and NLP, while VOLO is only used in image classification. We hope the reviewer to consider the superior performance of MHLA over all the tasks, instead of specializing in image classification.

---

> ### Author Response · Authors · 2025-11-26
> **Response to Reviewer WAkW (5)**
>
> **Q3.1: Asking for detailed settings**
>
> > The additional video generation experiments are interesting and potentially a strong point of the paper. However, implementation details are missing: e.g., training data, number of GPUs, training steps, and the exact evaluation benchmarks. Without these, it is difficult to properly assess the strength and practicality of the results.
> >
>
> We thank the reviewer for acknowledging our contribution.
>
> Our evaluation is conducted on **VBench [1]**, the most widely adopted benchmark for video generation. We trained the MHLA-enhanced WAN2.1-1.3B model on **800×480 resolution, 81-frame** data using **32×A100 GPUs** with a batch size of **128**. For the *Full MHLA* setting, we trained for **70k steps**, while *Hybrid 2/3* and *Hybrid 4/5* were trained for **15k steps**, which represents a relatively small training budget. We will include these detailed settings in the revised version of the paper.
>
> [1] VBench: Comprehensive Benchmark Suite for Video Generative Models, CVPR 2024
>
> **Q3.2: Is Vanilla Linear attention a weak baseline?**
>
> > Moreover, surpassing a “native” linear attention baseline in video generation is not difficult. Vanilla linear attention is known to be a relatively weak choice compared to more specialized efficient-attention.
> >
>
> We thank the reviewer for the suggestion. We would like to highlight that in image classification, image generation, and NLP, we have already compared MHLA against *stronger* linear-attention variants, and MHLA consistently outperforms them. For video generation, linear attention is still an emerging research field, and to the best of our knowledge, there are currently no other suitable peer-reviewed linear-attention-based video generation methods available for comparison. If the reviewer is aware of a strictly stronger linear-attention-based baseline for video generation, we would greatly appreciate it if the reviewer could provide a peer-reviewed paper as a reference with stronger performance.
>
> **Q3.3: Is Linear Attention a worse choice compared with Sparse Attention?**
>
> > Vanilla linear attention is known to be a relatively weak choice compared to more specialized efficient-attention or sparse-attention designs. If you want to claim video generation as a key contribution of MHLA, it would be much more convincing to compare against dedicated accelerated video generation methods, such as STA [1] or sparse video generation approaches [2,3,4], under comparable training budgets and evaluation protocols.
> >
>
> We thank the reviewer for the suggestion. We would like to emphasize that the baselines listed by the reviewer are **sparse softmax attention or sliding window-based softmax attention methods**, **which are orthogonal architecture modifications compared to** **linear attention**.
>
> However, to address the reviewer's concern, we provide the comparison result of MHLA with VSA [1] in the table below, and the results show that **MHLA-Hybrid achieves better overall VBench scores under comparable or even higher speedups**, while **Full MHLA provides substantially larger acceleration with a similar overall score**. These results consistently demonstrate the effectiveness of MHLA.
>
> |  | VBench Overall Score$\uparrow$ | speedup$\uparrow$ |
> | --- | --- | --- |
> | VSA [1]  | 82.77 | 1.72x (with torch.compile) |
> | 2/3 MHLA hybrid | 83.82 | 1.61x |
> | 4/5 MHLA hybird | 83.71 | 1.80x |
> | Full MHLA | 82.83 (± 0.21) | 2.05x |
>
>  [1] VSA: Faster Video Diffusion with Trainable Sparse Attention. NeurIPS 2025

---

> ### Author Response · Authors · 2025-11-30
> **Summary of Discussion with Reviewer WAkW**
>
> We have had a comprehensive discussion with Reviewer WAkW. We summarize some key concerns from the reviewer and how we responded. For the detailed explanation, please refer to the rebuttal process.
>
> 1. **Performance with the newly proposed image classification baseline, VOLO**
>
>     We have pointed out that the reviewer is using incorrect results for MHLA when comparing with VOLO. MHLA is 0.4% higher than VOLO in accuracy, with 30% less computation.
>
> 2. **Additional comparison with a larger model size**
>
>     As requested by the reviewer, we have trained a larger model to compare with VOLO, where MHLA achieves consistent superior performance over VOLO with 0.4% higher accuracy, 15% fewer parameters, and 30% less computation.
>
> 3. **Conceptual difference between MHLA and VOLO**
>
>     We have explained that MHLA is improving the linear attention’s performance, while VOLO is under the category of dynamic convolution.
>
> 4. **Naming convention of MHLA**
>
>     As required by the reviewer, we have clearly stated the concept of token-level multi-head linear attention. If necessary, we could also change the name.

---

### Author Response · Authors · 2025-11-23
**Summary of Responses (1/2)**

We sincerely thank all the reviewers for their constructive and insightful feedback. Below, we summarize the key clarifications and additional results provided in the rebuttal.

**(1) Performance on generative and classification tasks.**

We would like to highlight that MHLA is proposed to be a general operator across both computer vision, generative tasks, and NLP tasks, where MHLA delivers **strong and consistent improvements**, surpassing current state-of-the-art methods in each field. It is designed as a **general-purpose architectural component.** The main ablation experiments lie in image classification and image generation tasks.  We found that most of the feedbacks from the reviewer focus on the NLP experiments. Thus, we sincerely invite the reviewers to consider our demonstrated effectiveness beyond the NLP setting, given that we have carefully addressed all NLP-related questions and provided additional evidence as requested.

**(2) Performance under ultra-long sequence regimes.**

Several reviewers expressed interest in MHLA’s behavior at extreme sequence lengths. To this end, we conducted new experiments on **video generation**, where sequence lengths reach **32k tokens**. The results show that standard linear attention suffers from **severe performance degradation**, consistent with our observation of *global context collapse*. In contrast, MHLA converges smoothly and achieves **strong performance**, reinforcing its robustness in ultra-long contexts, while keeping the exact same computational complexity as vanilla linear attention.

**(3) Expanded evaluation on NLP benchmarks.**

In response to requests for more comprehensive NLP results, we conducted additional evaluations on **common-sense reasoning** benchmarks, including **MMLU** and **LongBench**, and report detailed results in the rebuttal. These experiments demonstrate MHLA’s **broad applicability** to language modeling and show that it achieves performance **comparable to leading SOTA models** across diverse tasks.

Once again, we thank the reviewers for their thoughtful feedback. We believe the additional experiments and clarifications adequately address all concerns, and we appreciate your consideration of our rebuttal.

Comparison on common-sense reasoning and MMLU:


| Tasks | MMLU acc $\uparrow$ | **CSR .avg score** $\uparrow$ | Wino acc $\uparrow$ | PIQA acc $\uparrow$ | ARC-c acc_n $\uparrow$ | OBQA acc_n $\uparrow$ | ARC-e acc_n $\uparrow$ | BoolQA acc $\uparrow$ | Wiki ppl $\downarrow$ | LMB ppl $\downarrow$ |
| --- | --- | --- | --- | --- | --- | --- | --- | --- | --- | --- |
| GLA (340M) | 22.9 | 46.0 | 50.0 | 62.9 | 25.5 | 31.0 | 45.8 | 60.8 | 41.47 | 86.98 |
| Transformer++(340M) | 22.9 | 46.8 | 49.6 | 64.4 | 25.7 | 32.8 | 48.1 | 60.5 | 34.57 | 60.46 |
| Mamba (390M) | 23.5 | 46.4 | 50.5 | 64.1 | 24.9 | 32.4 | **48.3** | 58.2 | 38.32 | 62.43 |
| Mamba2 (340M) | 23.0 | 47.0 | 49.8 | **64.6** | 25.5 | 32.0 | 49.2 | 61.2 | 35.40 | 58.51 |
| GDN (360M) | 23.0 | 46.9 | **51.3** | 64.5 | 25.4 | 31.4 | 47.3 | **62.0** | 35.01 | 60.16 |
| MHLA (340M) | **23.7** | **47.1** | **51.3** | 64.4 | **25.9** | **33.4** | 46.5 | 61.3 | 38.31 | 71.64 |

Comparison on LongBench:

|  | Overall Performance | Multi-Doc QA |  |  | Summarization |  |  | Code |  | Single-Doc QA |  | Few-shot |  | Synthetic Task |  |
| --- | --- | --- | --- | --- | --- | --- | --- | --- | --- | --- | --- | --- | --- | --- | --- |
| Model | avg $\uparrow$ | hotpotqa $\uparrow$ | musique $\uparrow$ | 2wikimqa $\uparrow$ | qmsum $\uparrow$ | gov_report $\uparrow$ | multi_news $\uparrow$ | repobench-p $\uparrow$ | lcc $\uparrow$ | qasper $\uparrow$ | nqa $\uparrow$ | samsum $\uparrow$ | triviqa $\uparrow$ | passage retrieval en $\uparrow$ | passage retrieval zh $\uparrow$ |
| Mamba | *6.97* | 2.36 | 1.60 | 3.37 | 12.23 | *18.36* | 14.96 | **13.63** | 12.33 | 4.57 | 2.28 | 5.16 | 5.49 | 1.1 | 0.1 |
| GLA | 6.53 | 2.31 | 1.67 | 3.23 | 11.42 | 17.72 | 15.34 | *13.59* | 12.55 | 4.53 | 2.13 | 3.94 | 0.7 | **1.98** | 0.27 |
| GDN | 6.86 | 2.24 | 1.54 | 2.86 | 12.46 | 17.91 | **15.98** | 10.42 | 9.98 | **4.73** | **2.48** | **6.85** | **7.61** | 0.53 | 0.41 |
| Transformer++ | 6.92 | **2.13** | **2.22** | **4.97** | 11.75 | 16.81 | *15.11* | 11.56 | 9.92 | 4.45 | 2.35 | 6.24 | *7.47* | 0.76 | 1.18 |
| Mamba2 | 6.62 | *2.38* | 1.69 | 3.56 | *12.57* | 17.65 | 14.00 | 10.15 | 9.49 | *4.70* | 2.20 | 4.97 | 7.03 | 0.72 | **1.51** |
| MHLA | **7.41** | **2.97** | *1.87* | *3.58* | **12.58** | **18.59** | 15.01 | 13.37 | **12.72** | 4.68 | *2.38* | *6.41* | 6.44 | *1.69* | *1.49* |

---

> ### Author Response · Authors · 2025-11-23
> **Summary of Responses (2/2)**
>
> Comparison on Video Generation:
>
> | Method | Quality score$\uparrow$ | Semantic score$\uparrow$ | Total$\uparrow$ | Latency$\downarrow$ |
> | --- | --- | --- | --- | --- |
> | Wan2.1 1.3B | 85.23 | 75.65 | 83.31 | 166s |
> | Full MHLA | 84.26 | 76.16 | 82.62 | 81s |
> | Full Linear  | 69.96 | 11.38 | 58.24 | 82s |
> | MHLA Hybrid 2/3 | 84.87 | 79.59 | 83.82 | 103s |
>
> *Note: “Full MHLA” and “Full Linear” indicate changing all flash attention to MHLA / vanilla linear attention. “MHLA Hybrid 2/3” means changing 2/3 layers to MHLA. Both MHLA and vanilla linear attention are accelerated with torch.compile.*

---

### Meta-Review · Area_Chair_T5Vg · 2026-01-02

**Summary:**

Reviewers broadly agreed that the paper makes a substantive and well-motivated contribution by identifying global context collapse as a key failure mode of linear attention and proposing Multi-Head Linear Attention (MHLA) as a principled remedy. The method was consistently viewed as insightful, and was regarded as simple, hardware-friendly, and broadly applicable across vision, generative modeling, and NLP. The concerns that initially informed hesitation or borderline scores focused on evaluation completeness, positioning, and practical clarity, rather than on correctness.

Overall, the work was seen as promising, but several reviewers felt that the claims around long-context effectiveness and efficiency required stronger empirical evidence and clearer articulation. The rebuttal substantially addressed these concerns with additional experiments and clarifications. I therefore recommend acceptance.

**Reviewer Concerns:**

The rebuttal substantially addressed the major technical and empirical concerns raised by reviewers:

Long-context and cross-domain validation: The authors added extensive new results on LongBench, MMLU, expanded commonsense reasoning benchmarks, video generation, and higher-resolution image classification.

Throughput, latency, and memory profiling: Detailed latency, throughput, and peak-memory measurements across vision, diffusion, and language models were added.

Scalability analysis: The rebuttal clarified the computational complexity, explicitly discussed how block size scales with sequence length, and empirically demonstrated stable performance at very long contexts.

Baseline coverage and fairness: Comparisons were expanded to include stronger linear/SSM baselines and additional diffusion baselines. For video generation, comparisons against sparse-attention methods were added.

Clarifications on VOLO and FLASH: The authors corrected factual inaccuracies in the initial VOLO comparison, added larger-model comparisons (e.g., MHLA vs. VOLO-D2), and clarified conceptual differences.

Statistical rigor and reporting: Multi-seed results and detailed evaluation protocols were provided where requested.

Terminology and exposition: The rebuttal clarified naming, explained the meaning of “token-level multi-head,” and justified new terms (e.g., “KV summary”) as reflecting mechanisms not present in prior linear attention.

**Reviewer Scores:**

Reviewer uTES: Initial score 6, kept 6 after rebuttal.
The reviewer acknowledged that concerns about evaluation breadth, long-context testing, and efficiency were addressed, but maintained a cautious stance.

Reviewer 2bkY: Initial score 4, increased to 6 after rebuttal.
The reviewer explicitly stated that the added long-context experiments and efficiency analyses satisfactorily addressed their concerns.

Reviewer WAkW: Initial score 2, may increase to 4 or 6 after rebuttal.
I think the authors have addressed the main concerns proposed by the reviewer.

---

### Decision · Program_Chairs · 2026-01-26

Accept (Poster)